**RESEARCH**

# Mapping the functional impact of non-coding regulatory elements in primary T cells through single-cell CRISPR screens

Celia Alda-Catalinas[1†], Ximena Ibarra-Soria[1†], Christina Flouri[1], Jorge Esparza Gordillo[1], Diana Cousminer[2], Anna Hutchinson[1], Bin Sun[1], William Pembroke[1], Sebastian Ullrich[1], Adam Krejci[3], Adrian Cortes[1], Alison Acevedo[1], Sunir Malla[1], Carl Fishwick[1], Gerard Drewes[1,2] and Radu Rapiteanu[1*]

†Celia Alda-Catalinas and Ximena Ibarra-Soria contributed equally to this work.

*Correspondence:
radu.a.rapiteanu@gsk.com

[1] Genomic Sciences, GSK, Stevenage, UK
[2] Genomic Sciences, GSK, Collegeville, PA, USA
[3] Myllia Biotechnology, Vienna, Austria

## Abstract

**Background:** Drug targets with genetic evidence are expected to increase clinical success by at least twofold. Yet, translating disease-associated genetic variants into functional knowledge remains a fundamental challenge of drug discovery. A key issue is that the vast majority of complex disease associations cannot be cleanly mapped to a gene. Immune disease-associated variants are enriched within regulatory elements found in T-cell-specific open chromatin regions.

**Results:** To identify genes and molecular programs modulated by these regulatory elements, we develop a CRISPRi-based single-cell functional screening approach in primary human T cells. Our pipeline enables the interrogation of transcriptomic changes induced by the perturbation of regulatory elements at scale. We first optimize an efficient CRISPRi protocol in primary CD4$^+$T cells via CROPseq vectors. Subsequently, we perform a screen targeting 45 non-coding regulatory elements and 35 transcription start sites and profile approximately 250,000 T-cell single-cell transcriptomes. We develop a bespoke analytical pipeline for element-to-gene (E2G) mapping and demonstrate that our method can identify both previously annotated and novel E2G links. Lastly, we integrate genetic association data for immune-related traits and demonstrate how our platform can aid in the identification of effector genes for GWAS loci.

**Conclusions:** We describe "primary T cell crisprQTL" — a scalable, single-cell functional genomics approach for mapping regulatory elements to genes in primary human T cells. We show how this framework can facilitate the interrogation of immune disease GWAS hits and propose that the combination of experimental and QTL-based techniques is likely to address the variant-to-function problem.

**Keywords:** Enhancer, Regulatory element, GWAS, Variant to function, crisprQTL, CRISPR, T cells, scRNA-seq, CROPseq

## Background

Genome-wide association studies (GWAS) have revealed thousands of disease-associated single-nucleotide polymorphisms (SNPs) [1]. Drug targets supported by human genetic evidence are expected to increase clinical success by at least twofold [2–6]. Thus, understanding the molecular mechanisms underpinning GWAS hits is key to reducing attrition in drug discovery. More than 90% of disease-associated variants are located in non-coding genomic regions [7–9], making it challenging to identify the causal effector gene(s) they regulate [10–14].

Non-coding disease-associated variants are enriched within cell type-specific open chromatin regions, especially regulatory elements such as promoters and enhancers [7, 11, 15–21], and they often impact gene expression in a cell type-specific manner [22–30]. Hence, several studies have combined genetic fine-mapping with epigenomic profiles to prioritize candidate *cis*-regulatory elements within trait-relevant cell populations [31–39]. However, identifying the genes and downstream molecular programs modulated by disease-associated regulatory elements remains difficult with currently available tools.

CD4$^+$ T cells play critical roles in autoimmune and inflammatory disorders, such as inflammatory bowel disease, type 1 diabetes, Crohn's disease, and rheumatoid arthritis [40, 41]. These cells are heterogeneous and highly plastic as they differentiate and acquire distinct functions to counter pathogens and navigate changing environments [42]. Fine-mapping of GWAS loci, expression quantitative trait loci (eQTLs), and epigenomic studies have shown that immune disease-associated risk variants are enriched in CD4$^+$ T-cell regulatory regions [18, 19, 22, 29, 31, 32, 35, 43, 44].

CRISPR is a powerful tool to functionally characterize and map non-coding regulatory elements to genes [45–52]. In recent years, the combination of CRISPR screening with single-cell RNA sequencing (scRNA-seq) has enabled deep phenotyping of genetic perturbations at scale [53–55], providing an unprecedented opportunity to disentangle genome regulation. Specifically, high-throughput single-cell CRISPR-interference (CRISPRi) screens of regulatory regions have been performed to generate enhancer-gene maps at scale [49, 52]. These studies have so far used immortalized cell lines due to their ease of manipulation. However, the epigenetic regulation of immortalized, highly passaged cell lines is adapted to their highly proliferative state rather than being representative of the tissues from which they were derived [56, 57]. Therefore, we sought to establish a method that allowed us to query the function of regulatory elements in a physiologically relevant cell context.

Here, we present the method called "primary T cell crisprQTL," a high-throughput, single-cell, pooled CRISPRi-based functional screening framework to map non-coding regulatory elements to genes in primary human CD4$^+$ T cells. We perturbed transcription start sites (TSSs) and putative regulatory elements with ZIM3-dCas9 and profiled 250,195 high-quality single CD4$^+$ T-cell transcriptomes. We developed an analytical pipeline to robustly assign perturbations to cells and to determine gene expression changes. We demonstrate that our method can identify high-confidence *cis* element-to-gene (E2G) pairs and nominate novel E2G links supported by genetic evidence.

## Results

### Implementation of single-cell CRISPRi technology in human primary CD4$^+$ T cells

To enable lentiviral-based CRISPRi perturbations in unexpanded primary CD4$^+$ T cells, we tested a panel of dCas9-KRAB lentiviral constructs targeting the TSS of four cell surface receptors: *CD4*, *CD81*, *BST2*, and *ATP1B3*. We designed dCas9 constructs using two KRAB repressor domains (KOX1 or ZIM3) under the control of two promoters (EFS or CBh) (Additional file 1: Fig. S1A). ZIM3 has been shown to improve CRISPRi silencing efficiency compared to the more widely used KOX1 domain [58]. Briefly, T cells were activated and transduced with dCas9-KRAB lentivirus and selected with blasticidin. Cells were then reactivated and transduced with TSS-targeting guide RNAs (gRNAs) or with a non-targeting (NT) control cloned into a CROPseq backbone [59]. After puromycin selection of cells expressing the gRNA, the effect of the CRISPRi perturbation on the expression of the target protein was analyzed by flow cytometry (Fig. 1A). We observed variable silencing efficacy for different genes (Fig. 1B, Additional file 1: Fig. S1B), likely due to differences in gRNA efficiency, basal gene expression, and local chromatin context, consistent with previous CRISPR modulation studies [60, 61]. Generally, using a ZIM3 repressor under a CBh promoter resulted in marginally improved silencing compared to the other constructs across multiple target genes (Additional file 1: Fig. S1B) and timepoints (Additional file 1: Fig. S1C). Thus, we selected the CBh-ZIM3-dCas9 construct for all subsequent experiments. To test the robustness of our method, we performed a CRISPRi experiment targeting the TSSs of *CD4*, *CD81*, and *BST2* in primary T cells derived from four donors, across a time-course series, and achieved up to 85% silencing efficiency (Fig. 1C).

Next, we assessed whether we could detect gRNA transcripts and quantify the downregulation of targeted genes by single-cell transcriptomics. Following independent transductions of gRNAs targeting the TSSs of *CD4*, *CD81*, *BST2*, and *ATP1B3*, we pooled all cells and performed a CROPseq experiment using 10X Genomics 3′ scRNA-seq. We were able to confidently assign gRNAs to cells and observed significant target gene downregulation compared to non-targeting controls (Fig. 1D, Additional file 1: Fig. S1D). Additionally, the magnitude of the expression changes measured by scRNA-seq was highly correlated with the protein abundance changes detected by flow cytometry (Additional file 1: Fig. S1E). These results show high-efficiency CRISPRi-based silencing of gene expression in primary CD4$^+$ T cells and demonstrate that perturbation effects can be read out at single-cell resolution using a CROPseq-based approach.

### Proof-of-concept crisprQTL screen recapitulates previously validated element-to-gene links

Having established the CRISPRi CROPseq workflow in primary T cells, we sought to use this pipeline for E2G mapping. We refer to this method as primary T cell crisprQTL. To show proof of concept, we silenced a panel of non-coding elements likely to regulate gene expression in primary T cells, alongside technical controls (Fig. 2A). First, we selected the *CD2* locus control region (LCR), which contains three regulatory elements that enhance *CD2* gene expression [62, 63]. Second, we identified 28 enhancers that overlap open chromatin in primary CD4$^+$ T cells and were previously paired to a gene

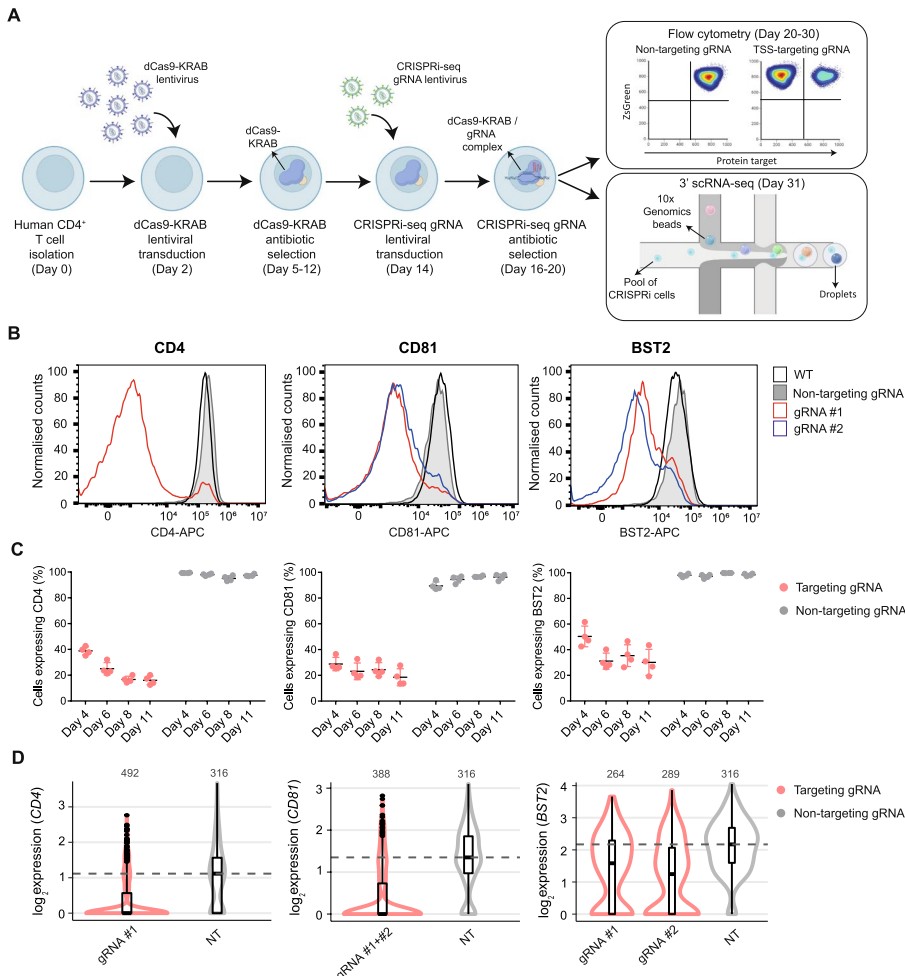

**Fig. 1** **A** Schematic of the CRISPRi protocol in primary CD4+ T cells. **B** Histograms showing expression of the target gene (*CD4*, *CD81*, *BST2*) 10 days after gRNA transduction into primary CD4+ T cells expressing a CBh-ZIM3-dCas9 repressor construct, analyzed by flow cytometry. gRNA #1 and gRNA #2 refer to two different gRNA designs for a given TSS. The wild-type (WT) control are non-transduced cells stained with the same antibody for the corresponding target gene. **C** Quantification of the percentage of cells retaining cell surface expression of *CD4*, *CD81*, and *BST2* at days 4, 6, 8, or 11 after transduction of a TSS-targeting gRNA (red) or NT control gRNA (gray) into primary CD4+ T cells expressing CBh-ZIM3-dCas9, analyzed by flow cytometry. Replicates are cells derived from four donors. Differences between non-targeting and targeting gRNAs are significant for all genes and timepoints (*p*-value < 0.00005, Bonferroni-Dunn test). **D** Normalized expression levels of the same target genes, measured by 10X Genomics 3′ scRNA-seq, 11 days after the corresponding targeting (red) or non-targeting (gray) gRNAs were transduced into primary CD4+ T cells expressing a CBh-ZIM3-dCas9 repressor construct. The dashed line indicates the median expression level in cells with non-targeting controls. The number of cells in each group is indicated at the top. Note gRNA #1 and #2 for *CD81* TSS were analyzed together due to sequence similarity

in a functional E2G mapping study in the K562 cell line (leukemia bone marrow-isolated lymphoblast cells) [49] (see "Materials and methods"); we refer to these elements as Gasperini enhancers (Gasperini_ENH). Third, we selected 14 non-coding elements from the ENCODE registry of candidate *cis*-regulatory elements (cCREs), 5 of which were intergenic, while the other 9 overlapped introns. Lastly, we included 35 TSSs of genes spanning a wide range of gene expression levels to serve as technical controls (Fig. 2A). For most of these perturbations, there is a predicted or *expected* downregulated gene

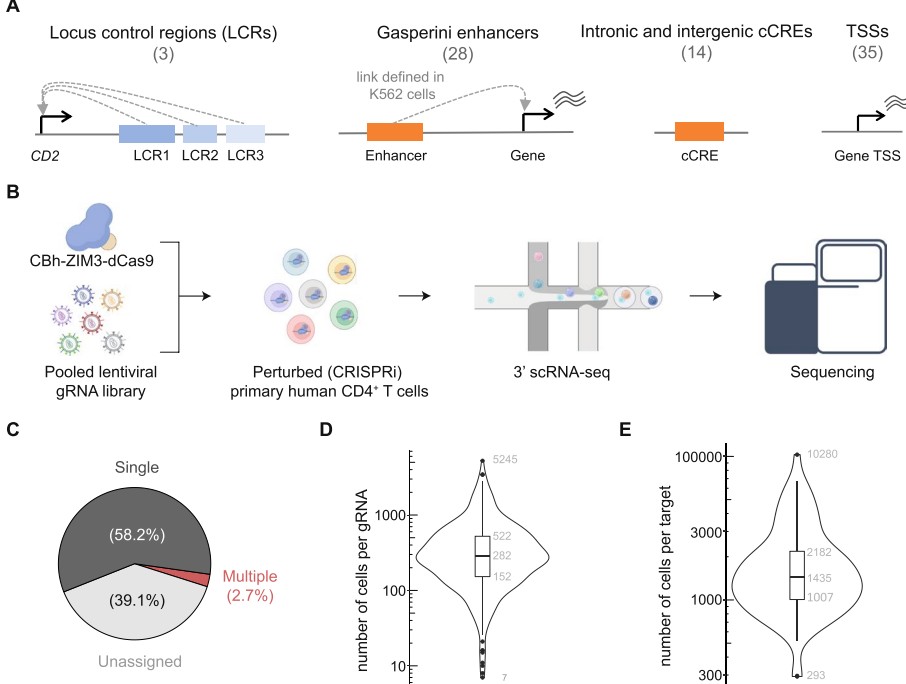

**Fig. 2 A** Schematic of the classes of loci targeted in the crisprQTL screen, including the locus control regions of *CD2*, enhancers linked to genes from Gasperini et al. [49], regulatory elements (intronic and intergenic) overlapping ENCODE cCREs, and gene transcription start sites (TSS). **B** Schematic of primary T cell crisprQTL experimental approach: CBh-ZIM3-dCas9 and the pooled gRNA library were introduced as described in Fig. 1A, and perturbed cells were analyzed by 10X Genomics 3′ scRNA-seq. **C** Proportion of cells where we confidently detected a single gRNA, multiple gRNAs, or none (unassigned, due to insufficient gRNA transcript recovery). **D** Distribution of the number of cells recovered with each gRNA in the pooled library. Numbers indicate, from top to bottom, the maximum, 75th, 50th, 25th quantiles, and minimum. **E** Same as **D** but for the number of cells per target (each target is targeted by four gRNAs)

based on previous evidence: *CD2* for LCR perturbations, the gene linked in K562 cells for enhancers selected from Gasperini et al. [49], and the gene immediately downstream of the TSS. Thus, we treated these perturbations as positive controls, whereas the cCRE perturbations were selected for exploratory E2G mapping.

We designed four gRNAs to target each candidate regulatory element or TSS along with 35 non-targeting controls (Additional file 2: Table S1) and cloned the resulting 355 gRNA library into our CRISPRi CROPseq backbone. This pooled gRNA library was transduced at low multiplicity of infection (MOI) into primary CD4$^+$ T cells previously selected for CBh-ZIM3-dCas9 expression. We generated high-quality transcriptional profiles and gRNA amplicon libraries for 250,195 single cells using the 3′ scRNA-seq 10X Genomics platform (Fig. 2B, Additional file 1: Fig. S2A).

To assign gRNAs to cells, we developed a probabilistic framework based on a binomial distribution to assess the likelihood that a gRNA is present in a cell, taking into consideration its representation in the initial plasmid library. Unlike the static thresholding methods previously employed [49, 64], we found that our framework allows us to control for two important sources of technical noise: biases in the efficiency of gRNA transcript recovery between cells and variation in the abundance of each gRNA species in the experiment, which influences the quantification noise from ambient RNA

and PCR artifacts. After applying this method to our gRNA amplicon data, we confidently assigned at least one gRNA to 152,403 (61%) cells (Fig. 2C, Additional file 1: Fig. S2B). Importantly, we verified that these gRNA assignments were consistent with gRNA transcripts recovered in the gene expression library, confirming that the PCR enrichment process does not introduce spurious signals (Additional file 1: Fig. S2C). As cells were transduced at low MOI, we identified a single gRNA in most cells (95.6% of cells assigned) (Fig. 2C). We recovered all 355 gRNAs present in the library, with a median of 282 cells per gRNA (Fig. 2D). All but 19 gRNAs were identified in more than 50 cells, and the numbers of cells assigned to each perturbation were correlated with the gRNA abundances in the plasmid library, indicating no significant adverse effects on cell fitness or viability (Additional file 1: Fig. S2D). The gRNAs showing poor recovery corresponded to a variety of targets. Thus, we achieved good representation of all perturbations in the experiment (median of 1,435 cells per target; Fig. 2E).

To determine the effects of each perturbation on the expression of nearby genes (1 Mb up and downstream from the targeting site), we used MAST, an algorithm for scRNA-seq differential expression analysis [65]. We first looked at the *expected* gene in the positive control perturbations and excluded two TSS and two Gasperini enhancer-positive control targets with poor expression (detection in fewer than 5% of cells). Cells transduced with gRNAs targeting a TSS showed significant ($FDR < 5\%$) downregulation for 32 out of the 33 targets (Fig. 3A–C), with an average reduction of 31% from wild-type expression levels (effect sizes in the range of 3–97%; Fig. 3B). Similarly, *CD2* expression was significantly downregulated upon silencing of any of the three LCR regions targeted (average 29% reduction; Fig. 3A–C). For the E2G pairs identified by Gasperini et al. in K562 cells [49] (Gasperini_ENH), we reproduced 17 out of the 26 associations in our primary T cell data, with an average reduction in gene expression levels of 20% (range 5–50%; Fig. 3A–C).

For gRNAs detected in at least 30 cells, we observed two major limitations in our ability to detect significant gene expression changes: the magnitude of the effect and the expression level of the affected gene (Fig. 3B, D). Since silencing the promoter results in strong downregulation of gene expression, we were able to detect the effects from TSS perturbations for genes covering the whole dynamic range of expression levels detected by single-cell transcriptomics (Fig. 3D). In contrast, silencing non-coding elements that affect genes expressed at lower levels did not reach statistical significance (Fig. 3D). Importantly, targets with significant effects were supported by at least two different gRNAs in 83% of cases, for both promoter and non-coding elements (Fig. 3A). Altogether, these results demonstrate that our crisprQTL method efficiently silences both gene promoters and distal regulatory elements, and that the effects of these perturbations on gene expression can be detected by single-cell transcriptomics, enabling the mapping of regulatory elements to genes in primary human CD4$^+$ T cells.

### Evaluation of analytical methods for high-confidence *cis* E2G mapping using crisprQTL

Having shown proof of concept based on downregulation of the *expected* gene for positive control perturbations, we analyzed expression changes across all genes within 1 Mb upstream and downstream of the targeted loci. Most positive control perturbations (86%) resulted in at least one additional significantly differentially expressed gene (DEG).

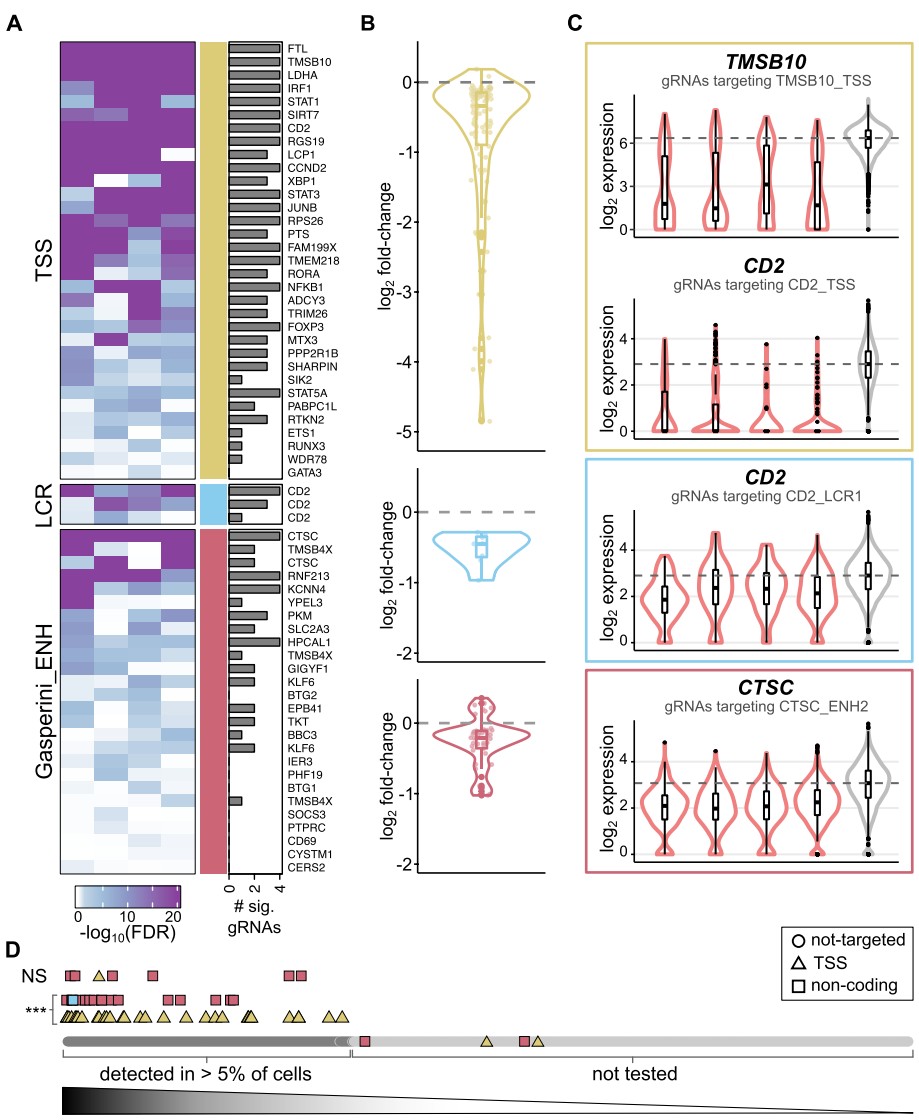

**Fig. 3 A** Heatmap of the differential expression significance values (-$\log_{10}$ adjusted *p*-value) for each of the four gRNAs targeting each of the positive control perturbations, when comparing the expression of the expected gene in perturbed cells versus non-targeting controls. Different classes of targets are indicated by colored bars (TSS — yellow, LCR — blue and Gasperini enhancers — red). The barplots to the right indicate how many of the four gRNAs reach statistical significance. **B** Distributions of the $\log_2$ fold-change values for all expected genes from positive control perturbations, split by target class. **C** Representative examples of targets from each class. Normalized expression values in cells with targeting gRNAs (red) versus NT controls (gray) are shown. The title of the plot indicates the gene plotted. **D** Plot depicting the effect of gene expression levels on our ability to detect downregulation effects upon perturbation of TSS and non-coding targets. At the bottom, all genes in the human genome are ranked by decreasing average expression in the scRNA-seq dataset. Only genes detected in at least 5% of the cells (dark gray) were considered in the differential expression analyses. Non tested genes (light gray) include both genes not expressed in T cells and genes not detected by scRNA-seq. Then, *expected* genes in positive control perturbations that were significantly differentially expressed (***) are indicated, separately for TSS (yellow triangles) and non-coding control perturbations (red squares for Gasperini_ENH target genes, blue square for *CD2*). Above, *expected* genes that were detected but not recovered as significantly downregulated upon perturbation (NS)

However, while *expected* genes were recovered with two or more independent gRNAs in most cases (83%) (Fig. 3A, Additional file 1: Fig. S3A), over two-thirds (68.3%) of all additional DEGs were statistically significant with a single gRNA (Additional file 1: Fig. S3A). This lack of reproducibility between different gRNAs for the same target could be the result of variable on-target efficiencies, off-target effects, and/or a sign of calibration issues of the statistical test. It is well recognized that differential gene expression testing using single-cell data results in the loss of appropriate false discovery rate (*FDR*) control when the lack of independence between cells from the same sample is not accounted for [66]. To assess whether this is the case for our data, we performed differential gene expression analysis for cells bearing non-targeting gRNAs, whose expression should not result in significant DEGs. Indeed, we observed that *p*-values were overly significant (Additional file 1: Fig. S3B), resulting in an excess of positive DEG calls.

To assess whether this is a problem specific to MAST, we tested two other differential gene expression algorithms, limma-voom [67] and SCEPTRE [68], along with a nonparametric test (Wilcoxon rank-sum test). All algorithms produced inflated *p*-values (Additional file 1: Fig. S3C). When comparing the significant gRNA-DEG pairs reported by each method, we observed that a large fraction was only identified by one of the four methods (Additional file 1: Fig. S3D); these corresponded almost entirely (94.2%) to genes other than the *expected* DEG, supporting that these are likely false positives. Instead, the *expected* genes from positive control perturbations were consistently detected by all four methods (Additional file 1: Fig. S3D), building confidence in our experimental assay. Additionally, we observed that gRNA-DEG pairs reported by two or more methods were frequently supported by two or more independent gRNAs, while the method-specific pairs were most often identified with a single gRNA (Additional file 1: Fig. S3E).

From all the methods considered, MAST showed the best recovery of *expected* gene expression changes along with the fewest method-specific pairs. Thus, we decided to use MAST results for all downstream analyses. To increase confidence in this set of DEGs, we exploited the evidence provided by independent gRNAs targeting the same element. We used Fisher's method to integrate the *p*-values from all four gRNAs from each target into a single statistic — a *target-level p-value* — which reflects the amount of evidence supporting a gene expression change in perturbed cells (Fig. 4A). After adjusting for the multiple tests performed for the whole library, we obtained 378 significant DEGs (*FDR* < 5%; Additional file 3: Table S2). We further classified these DEGs into confidence tiers, based on how many gRNAs supported the expression change (see "Materials and methods"). Altogether, we identified 87 high-confidence expression changes supported by 3 or 4 gRNAs for a given target, 140 medium-confidence DEGs supported by 2 gRNAs, and 151 low-confidence effects supported by a single gRNA (Fig. 4A, Additional file 3: Table S2). Since expression differences observed with a single gRNA are more likely to be off-target effects and/or false positives, we focused our analyses only on medium and high-confidence results.

For perturbations targeting Gasperini enhancers and cCREs (Fig. 2A), we detected 94 E2G pairs. These included at least one significant DEG for 22 of the 28 Gasperini perturbations, 9 of the 11 cCREs overlapping introns, and for all three cCREs in intergenic loci (Additional file 3: Table S2). In general, non-coding perturbations did not induce

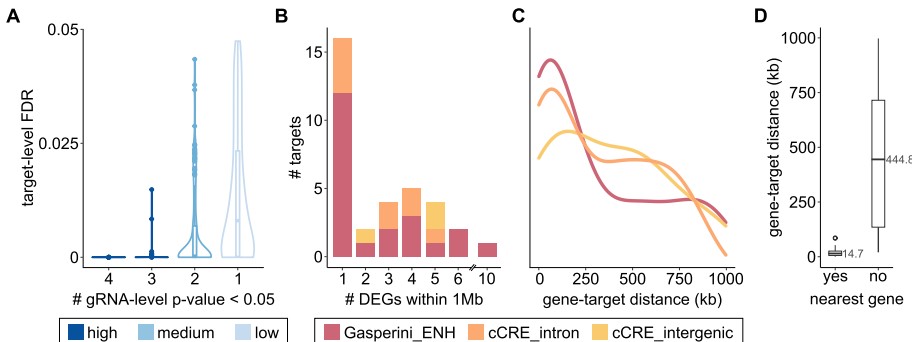

**Fig. 4 A** Distribution of the target-level adjusted *p*-values (FDR) for all significant element-to-gene (E2G) pairs detected, split by how many gRNAs have raw gRNA-level *p*-values < 0.05. E2G pairs supported by three or four gRNAs are high confidence (dark blue), E2G pairs supported by 2 gRNAs are medium confidence (blue), and E2G pairs supported by a single gRNA are low confidence (light blue) and were discarded from downstream analyses. **B** Barplot indicating the number of high and medium confidence significant differentially expressed genes (DEGs) detected for non-coding perturbations within 1 Mb up/downstream of the target site. Targets from Gasperini enhancers are shown in red; ENCODE cCREs are shown in orange if they lie within a gene intron and in yellow if they are intergenic. **C** Density plot of the distance between the E2G pairs, in kilobases. DEGs from targets of different classes are shown separately, as indicated by the same colors used in **B**. **D** Boxplots of the distance between E2G pairs (as in **C**) but split by whether the gene is the nearest expressed gene to the target. The median is indicated

widespread changes in expression. Almost half of all perturbations (47%) resulted in a single DEG within 1 Mb of the target site, and an additional 32% resulted in fewer than five DEGs (Fig. 4B). These results suggest that CRISPRi effects are specific to the targeted elements. The majority (70.1%) of these non-coding element perturbations resulted in dysregulation of the nearest expressed gene, located a median distance of 15 kb away from the target site (interquartile range (IQR): 5.5–25.8 kb; Fig. 4C–D). However, we also identified long-range effects affecting other genes, with a median distance of 445 kb (*IQR* IQR135–715 kb; Fig. 4C–D).

**crisprQTL E2G links are validated using an orthogonal approach**

We next sought to validate the screen results by inducing targeted element deletions coupled to bulk RNA-seq. We selected four E2G pairs across various gene expression levels and magnitudes of perturbation-induced effects (Fig. 5A). gRNA/Cas9-nuclease ribonucleoprotein complexes (RNPs) were used to induce element deletions in primary CD4[+] T cells derived from two independent donors (Fig. 5B). Deletions were confirmed by PCR and automated electrophoresis (Fig. 5C). Next, we analyzed the perturbation-induced transcriptomic changes by bulk RNA-seq. Differential expression analysis (Additional file 4: Table S3) confirmed significant downregulation of the same gene observed by crisprQTL in three out of the four cases (Fig. 5D). In the fourth case, upon deletion of KCNN4_ENH, we observed lower *KCNN4* gene expression in cells derived from both donors; however, the effect did not reach statistical significance. Taken together, these data corroborate the results of the crisprQTL screen.

**Primary T cell crisprQTL helps interrogate disease-associated loci**

Next, we focused on three case studies to emphasize how crisprQTL can aid the identification of effector genes for disease GWAS-associated regions. First, where expression

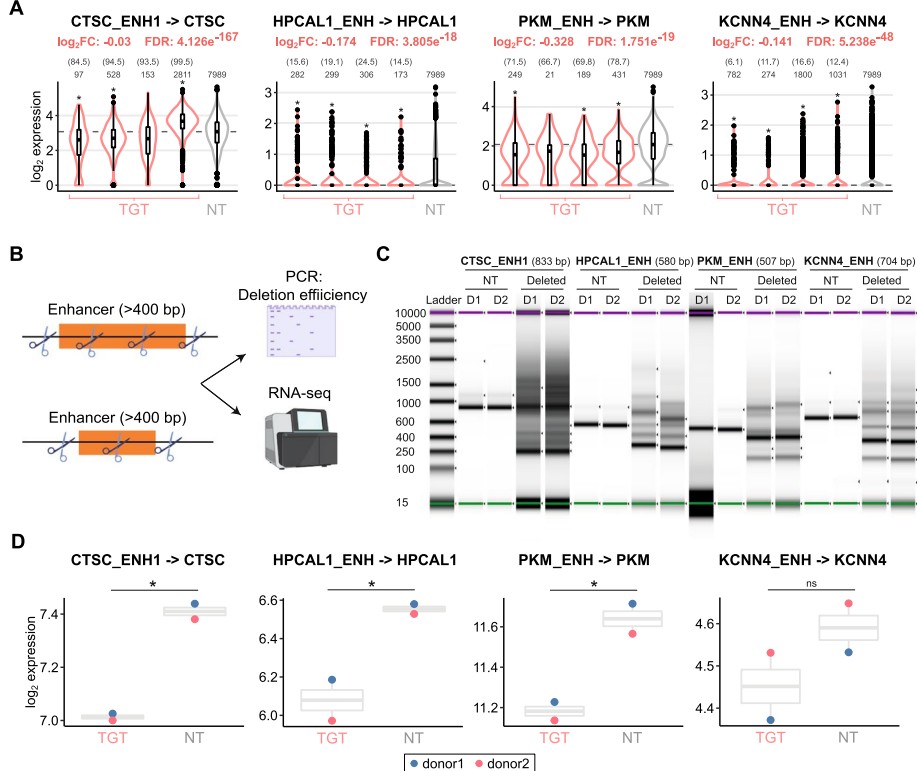

**Fig. 5 A** Normalized expression values in the crisprQTL screen for cells with targeting (TGT) gRNAs (red) versus non-targeting (NT) controls (gray) for the four E2G links selected for orthogonal validation. **B** Schematic of experimental approach used to induce targeted element deletions in primary CD4.[+] T cells, using gRNA/Cas9-nuclease ribonucleoprotein complexes (RNPs). The efficiency of the deletions was analyzed by PCR and automated electrophoresis. The perturbation-induced transcriptomic changes were assessed by bulk RNA-seq. **C** Automated electrophoresis analysis via TapeStation of the PCR products obtained after amplifying the targeted region in non-targeting (NT) control samples and CRISPR-deleted samples for four enhancer elements, across two donors (D1: donor 1; D2: donor 2). The size of the expected wild-type band is shown in brackets for each enhancer perturbation. **D** Normalized expression values for the DEG identified in the crisprQTL screen (**A**) in bulk RNA-seq data from cells with CRISPR deletion of the corresponding enhancer or with a NT control, across two donors. All four genes show the expected downregulation of expression, and three reach statistical significance (*adjusted *p*-value < 0.05)

data in the relevant cell type is publicly available, our crisprQTL data replicates colocalizations between disease GWAS signals and eQTL signals. For example, when perturbing the enhancer linked to *GIGYF1*, which overlaps SNPs in the 95% credible set for a type 2 diabetes (T2D) GWAS signal (Fig. 6A) [69], we found that *GIGYF1* was significantly differentially expressed ($FDR = 5.14\mathrm{e}^{-14}$), with perturbed cells showing approximately a 10% reduction in expression (Fig. 6B). Furthermore, the T2D GWAS signal colocalized with an eQTL signal for *GIGYF1* in CD4[+] T cells (Fig. 6C) [33, 70] and increased T2D risk colocalized with lower *GIGYF1* expression (Fig. 6D).

Second, for some enhancers, we identified E2G pairs in T cells that were different from those reported by Gasperini et al. (2019) in K562 cells [49], suggesting cell-type-specific regulation. For instance, the perturbation of an enhancer element upstream of *PHF19* (Fig. 6E) that overlaps SNPs in the 95% credible set for a rheumatoid arthritis GWAS signal [71] resulted in immune-related DEGs not reported in K562 cells [49], including *TRAF1* (Fig. 6F). This rheumatoid arthritis GWAS signal colocalized with a *TRAF1*

eQTL signal in CD4[+] T cells [24, 72] (Fig. 6G) and increased risk for rheumatoid arthritis colocalized with increased expression of *TRAF1* (Fig. 6H). This exemplifies how crisprQTL can help resolve cell-type-specific putative effector genes at GWAS loci.

Finally, our method can also be used to identify novel E2G links that are missed with other approaches. For example, when perturbing an intergenic element upstream of *CXCR5*, we detected differential expression of *CD3D*, whose TSS is more than 500 kb away from the perturbed region (Fig. 6I–J). The protein encoded by *CD3D* is a key component of the CD3 T-cell co-receptor that plays essential roles in the adaptive immune response. The target perturbed region overlaps credible set SNPs for various immune-related traits, including asthma [73], multiple sclerosis [74], atopic dermatitis [75], rheumatoid arthritis [71], and rhinitis [73] (Fig. 6I). We found no eQTL colocalization evidence supporting this E2G link. Together, our data suggest that crisprQTL can identify and validate cell-type-specific E2G links supported by genetic evidence as well as discover novel associations that are missed in population-based studies.

## Discussion

We present a novel methodology called primary T cell crisprQTL — a CRISPRi-based single-cell functional screening approach in primary human CD4[+] T cells that enables the mapping of regulatory elements to effector genes. Thus far, GWAS has identified a plethora of unique immune disease associations, many of which reside in cCREs in T cells [18, 19, 22, 29, 31, 32, 35, 43, 44]. Understanding the molecular mechanisms governed by these immune disease-associated regulatory elements would aid the understanding of disease etiology and pave the way towards novel therapeutics [2, 5]. The primary challenge, however, is identifying effector genes. Traditionally, eQTL studies have been used for this purpose, but eQTLs only clarify the effector gene at a limited fraction of GWAS signals [12, 76–80]. Furthermore, systematic studies have shown

(See figure on next page.)

**Fig. 6** **A** GWAS regional association plot for type 2 diabetes (T2D) [69] highlighting the perturbed region with the gray line near *GIGYF1*. **B** On the left, violin plots showing the normalized expression values of *GIGYF1* in cells expressing *GIGYF1* enhancer targeting (TGT) gRNAs (red) versus non-targeting (NT) controls (gray). The number of cells in each group is indicated at the top of the violins. On the right, barplot indicating the proportion of cells with TGT or NT gRNAs where expression of GIGYF1 is detected (counts > 0). The number of cells in each group is indicated at the top, and an * indicates the perturbation was significant at the gRNA level. The target-level corrected *p*-value (FDR) of expression change and a summary log$_2$ fold-change are indicated at the top. **C** eQTL regional association plot for *GIGYF1* expression in naïve CD4[+] T cells [33, 70], highlighting the perturbed region in a gray line. **D** Colocalization plot of the T2D GWAS signal (**A**) and *GIGYF1* eQTL in naïve CD4[+] T cells (**C**), showing that these signals have a 99% posterior probability of being shared. T2D risk colocalizes with decreased *GIGYF1* transcript expression. **E** GWAS regional association plot for rheumatoid arthritis (RA) [71] highlighting the perturbed region in gray near *PHF19* and *TRAF1*. **F** Same as **B** but for expression of *TRAF1* in cells expressing *PHF19* enhancer targeting (TGT) gRNAs (red) versus NT controls (gray). **G** eQTL regional association plot for *TRAF1* expression in naïve CD4[+] T cells, highlighting the perturbed region with a gray line [24, 72]. **H** Colocalization of the RA GWAS signal (**E**) and *TRAF1* eQTL in naïve CD4[+] T cells (**G**), showing that these signals have an 87% posterior probability of being shared. RA risk colocalizes with increased *TRAF1* transcript expression. **I** GWAS regional association plot for allergic and chronic rhinitis [73] highlighting the perturbed area with a gray line near *CXCR5*. **J** Violin plots showing the normalized expression values of *CD3D* in cells expressing targeting (TGT) gRNAs for *CXCR5* intergenic element (red) versus NT controls (gray). The number of cells in each group is indicated at the top, and an * indicates the perturbation was significant at the gRNA level. The target-level corrected *p*-value (FDR) of expression change and a summary log$_2$ fold change are indicated at the top

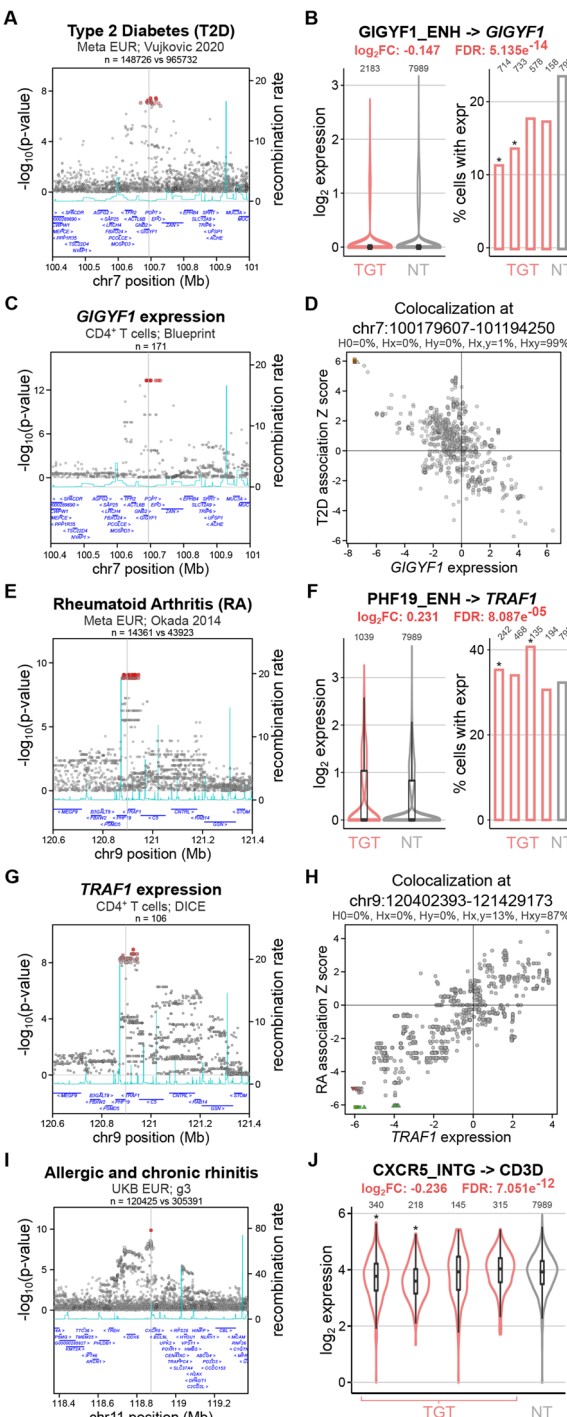

**Fig. 6** (See legend on previous page.)

that GWAS and eQTL methods are biased to identify different types of variants, with eQTLs being depleted at functionally important variants [79]. Therefore, the use of functional methodologies, such as crisprQTL, provides a complementary approach for gene mapping.

We propose crisprQTL in disease-relevant cell models as a powerful functional genomics tool to help refine effector gene mapping at genetically supported target loci. In this study, we defined bona fide controls to enable future large-scale crisprQTL screens aiming to interrogate immune disease-associated elements in primary T cells. Beyond TSSs and locus control regions, we selected enhancer-element pairs previously identified in the K562 chronic myelogenous leukemia cell line [49] and applied filters to ensure the following: (1) the elements overlapped open chromatin in primary T cells, and (2) the gene was expressed in primary T cells. This approach can be applied for the development of crisprQTL studies in further disease relevant models. Out of the 28 prioritized enhancers, 17 mapped to the same effector gene in T cells and K562 cells, indicating conserved function. However, 12 showed distinct and/or additional gene targets, indicating cell-type-specific regulation. For example, we perturbed an enhancer upstream of *PHF19* that was mapped to *PHF19* in crisprQTL studies in K562 cells [49] but to *TRAF1* in our primary T cell crisprQTL screen. Our observation was corroborated by colocalization of a rheumatoid arthritis GWAS signal overlapping the enhancer with a *TRAF1* eQTL signal in CD4$^+$ T cells (Fig. 6E–H). These findings highlight the importance of applying functional E2G mapping approaches in relevant cell models.

Recent advances in single-cell protocols, together with a drop in sequencing costs, are facilitating the generation of larger datasets querying cells derived from multiple donors, up to whole patient cohorts. The inclusion of higher biological replication will limit false discoveries and improve power to detect subtle perturbation effects [81]. Furthermore, the single-cell resolution inherent to crisprQTL enables the study of the impact of cell subtypes and states on the complex interrelationships between genetic variation, regulatory elements, and genes. This will provide a valuable resource for higher-throughput crisprQTL screens in primary, dynamic, and complex cell models and will enable delineating cell subtype- or state-specific regulation driven by disease-associated variants [22–30, 82–84]. For example, studies modelling eQTLs obtained from single-cell RNA-seq data from primary T cells helped disentangle cell subtype-specific effects that were masked in bulk RNA-seq [84]. Others have identified loci with independent eQTLs that have opposing state-dependent effects [83]. Furthermore, using H3K27ac HiChIP data in primary human naïve T cells, regulatory T cells (Treg), and T helper 17 cells (Th17), Mumbach et al. (2017) identified links between autoimmune disease variants and effector genes that were present in T effector cell types (Treg and Th17) but not in naïve T cells [22].

To enable the perturbation of regulatory elements, we established CRISPRi in primary CD4$^+$ T cells by lentiviral transduction of both dCas9-repressor and pooled gRNA library. A similar approach [85] was used for gene silencing in primary T cells using the KOX1 KRAB domain, to enable fluorescence-activated cell sorting (FACS)-based screens. We envision that our CRISPRi-based E2G methodology can be applied to other cell types and tissues amenable to viral transduction. However, this will most likely require extensive optimization of transduction conditions, including testing of viral pseudotypes that can facilitate efficient infection of the desired cell type. Similarly, we found that gRNA detection in scRNA-seq libraries of primary T cells can be particularly challenging, and thus, we anticipate it will require systematic optimization in other primary cell models.

While we and others have shown that pooled CRISPRi screens coupled to scRNA-seq enable unbiased interrogation of hundreds of elements in a single assay [49, 52], other perturbation methods for the discovery and mechanistic study of E2G associations are rapidly evolving and could provide orthogonal means of validation. For example, we used targeted CRISPR deletions to validate four E2G links discovered in our screen. Others have employed CRISPR/Cas9-directed mutagenesis to interrogate non-coding regulatory elements at defined loci [86–92]. Although these methods suffer from limited scalability, their application to prioritized loci could uncover regulatory maps at much higher precision. CRISPR-activation and other CRISPR epigenetic editing methods have also been employed to study non-coding regulation in small-scale studies [45, 93, 94]. Moving forward, a systematic comparison and benchmarking of E2G perturbation maps generated with multiple CRISPR modifiers and precise edits across multiple tissues should help us define robust principles for E2G interrogation and discovery.

## Conclusions

We present "primary T cell crisprQTL" — a single-cell functional framework for mapping disease-associated elements to genes in primary cells. We identify previously annotated and novel element-to-gene (E2G) links and further validate four E2G links using an orthogonal approach. We highlight three case studies to emphasize how crisprQTL can aid the identification of effector genes for immune disease GWAS hits. Our experimental perturbation approach coupled to a bespoke analytical pipeline sets the basis for future functional interrogation of immune disease-associated distal elements at scale. We firmly believe that, together with population-based studies, crisprQTL E2G maps across multiple human tissues will aid our understanding of the effector genes and pathways driving human disease.

## Methods

### Locus selection

The 35 TSSs were selected based on the following criteria (Additional file 2: Table S1): (1) 15 genes with essential roles in T-cell regulation and (2) 20 genes selected based on expression, as assessed from a scRNA-seq dataset from CD3/CD28 stimulated CD4$^+$ T cells [95] (genes were binned into quartiles based on average expression: 11 highly expressed genes were selected from the top quartile, 6 from the 3rd expression quartile, and 3 from the lowest two expression quartiles).

The genomic coordinates for the three *CD2* LCRs were obtained from [62, 63, 96] (Additional file 2: Table S1).

We selected enhancers from the set of 470 high-confidence intergenic E2G pairs from Gasperini et al. [49] with the following features: (1) the putative enhancer must overlap a region of open chromatin in primary CD4$^+$ T cells as identified in at least one of the following datasets: (i) ATAC-seq peaks identified on CD4$^+$ T naïve or memory cells after 16 h or 5 days of CD3/CD28 stimulation [35] or (ii) ATAC-seq peaks from multiple CD4$^+$ T cells from [19]; (2) the *expected* target gene, as defined by Gasperini et al. [49], was amongst the top 50% expressed genes in primary T cells, as assessed from a scRNA-seq dataset of CD3/CD28 stimulated CD4$^+$ T cells [95]. After applying these filters, a set of 28 E2G pairs were selected to include in the screen (Additional file 2: Table S1).

Additionally, we selected 11 intronic and 3 intergenic cCREs [15] overlapping open chromatin regions in primary T cells [19] (Additional file 2: Table S1).

### CRISPRi gRNA design

Four gRNAs were included per target in the pooled library. gRNAs targeting TSSs were selected from the Dolcetto library [97]. gRNAs targeting non-coding elements (*CD2* LCRs, enhancers selected from Gasperini et al. (2019) [49], and ENCODE cCREs) were designed using the Broad Institute platform *CRISPick* [97] and selected to target the middle 300-bp region of the element (Additional file 2: Table S1). For enhancers selected from Gasperini et al. (2019), two out of the four gRNAs overlapped the gRNAs used in the original study by Gasperini et al. (2019) [49]. Lastly, 35 non-targeting gRNAs were included in the pooled library, selected from the Dolcetto library [97]. gRNA sequences for each element are reported in Additional file 2: Table S1.

### gRNA cloning and pooled gRNA library construction

Individual gRNAs and the pooled library were cloned into a CROPseq lentiviral backbone adapted from [59] with the following modifications: (1) the scaffold sequence was modified to include the optimized CRISPRi scaffold described in [98], and (2) a fluorescent ZsGreen marker was included downstream of the EF1α promoter and linked through T2A to a puromycin resistance cassette. This backbone is referred to as CRISPRi CROPseq.

#### Cloning of individual gRNAs

To clone gRNAs into this backbone (GENEWIZ), oligos containing the protospacer sequence and recombination arms homologous to the vector backbone were synthesized (oligo structure: 5′ tttcttggctttatatatcttgtggaaaggacgaaacacc-protospacer-gtttaagagctatgctggaaacagcatagcaagtttaaat 3′). The CRISPRi-seq backbone was digested with AarI and then cloned with the insert by recombination-based cloning. The reaction mixture was transformed into DH5α competent cells, and the colonies were selected at 37 °C. Individual colonies were picked, and plasmid DNA was isolated and verified by Sanger sequencing. The sequences of TSS-targeting gRNAs and controls used in Fig. 1 and Additional file 1: Fig. S1 are detailed in the following table:

| gRNA ID | Target | gRNA sequence |
| --- | --- | --- |
| *CD4* gRNA_1 | *CD4* TSS | AACAAAGCACCCTCCCCACT |
| *CD81* gRNA_1 | *CD81* TSS | GCCTGGCAGGATGCGCGGTG |
| *CD81* gRNA_2 | *CD81* TSS | GGCCTGGCAGGATGCGCGGT |
| *BST2* gRNA_1 | *BST2* TSS | CAGAGTGCCCATGGAAGACG |
| *BST2* gRNA_2 | *BST2* TSS | CGCTTATCCCCGTCTTCCAT |
| *ATP1B3* gRNA_1 | *ATP1B3* TSS | GAGTACTCCCCGTAACGAGG |
| *ATP1B3* gRNA_2 | *ATP1B3* TSS | GACGGCAGTGAAGGGTGGGA |
| NT gRNA | NA | AAAACAGGACGATGTGCGGC |

#### Cloning of pooled library (355 gRNAs)

gRNA oligos were synthesized, PCR amplified, and cloned into the CRISPRi CROPseq backbone using Gibson assembly, as previously described [99], by VectorBuilder. The

pooled gRNA library was built from a total of $7 \times 10^5$ single colonies, which represents more than 1000-fold coverage of the designed gRNAs. Library gRNA representation was assessed by 150-bp paired-end sequencing (Illumina, NovaSeq). The distribution of gRNA representation is shown in Additional file 1: Fig. S2D.

### Lentiviral vector production and determination of viral titer

HEK293T suspension-adapted cells (in-house) were cultured in growth media consisting of BalanCD HEK293 medium (Irvine Scientific, 91165) supplemented with 2% GlutaMAX (Thermo Fisher Scientific, 35050061) and 1% Pluronic F-68 (Thermo Fisher Scientific, 24040032). For lentiviral packaging, cells were seeded at $2 \times 10^6$ cells/ml density and transfected with a total 190 μg DNA complexed with 450 μg PEIpro (Polyplus-transfection, 10100,033). The plasmid DNA mix, which included an envelope encoding *VSVg*, two packaging plasmids encoding *rev*, *gag* and *pol* genes, and a transfer DNA plasmid vector, was added to pre-warmed OptiMEM (Thermo Fisher Scientific, 31985070), followed by addition of PEIpro and incubation for 30 min at room temperature for the transfection complexes to form. The transfection mixture was added to the HEK293T suspension adapted cells, and the cells were incubated in a shaker cell culture incubator (Multitron shaker incubator, Infors HT) at 37 °C with 5% $CO_2$ at 110 rpm. The next day, 1.5 mM sodium butyrate (Sigma, 303410) was added to the transfected cell culture. Lentiviral vector-containing media from the transfected cells was collected 72 h after transfection, clarified by centrifugation at $500 \times g$ for 10 min, filtered through 0.45 μm filter units, and concentrated via high-speed centrifugation at $70,000 \times g$ for 2 h. The viral pellet was then resuspended in RPMI media (Gibco, 11534446) at a $\sim 350 \times$ concentration. The resuspended lentiviral vector solution was aliquoted and stored at $-80$ °C.

Functional viral titer was determined by transducing $1 \times 10^5$ HEK293T adherent cells (Lenti-X™ 293 T Cell Line, 632180) with serial dilutions of the lentiviral solution in a total volume of 100 μL per well of a 96-well plate and incubated at 37 °C in 5% $CO_2$. For lentiviral vectors containing the fluorescent marker ZsGreen, including the pooled gRNA library, the titration was also performed in primary CD4$^+$ T cells, and the percentage of cells expressing ZsGreen was quantified 2 to 3 days post-transduction by flow cytometry using CytoFLEX S (Beckman) flow cytometer and normalized to the starting cell number and dilution factor to obtain the titer in transduction units per mL (TU/mL).

### Isolation and culture of primary human CD4$^+$ T cells

The human biological samples used in this study were sourced ethically, and their research use was in accord with the terms of the informed consent under an IRB/EC-approved protocol (IRB approval number 20190318) and reviewed by the WIRB (Western Institutional Review Board). Briefly, mononuclear cells from circulating blood were removed by apheresis. From these leukopaks, primary human CD4$^+$ T cells were enriched using CliniMACS Prodigy setup and human CD4 MicroBeads (Miltenyi, 130–045-101). Following positive selection, cells were aliquoted and cryopreserved. Primary CD4$^+$ T-cell vials were thawed at the time of the experiments and cells cultured in RPMI 1640 (Gibco, 11340892) supplemented with 10% heat-inactivated FBS, $1 \times$ GlutaMAX (Thermo Fisher Scientific, 35050061), 1 mM sodium

pyruvate (Thermo Fisher Scientific, 11360088), 5 mM HEPES (Thermo Fisher Scientific, 15630080), $1 \times$ nonessential amino acids (Thermo Fisher Scientific, 11140035), $1 \times$ penicillin/streptomycin (Thermo Fisher Scientific, 15070063), 55 µM 2-mercaptoethanol (Thermo Fisher Scientific, 21985023), and 15 ng/mL of recombinant human interleukin 2 (IL-2) (PeproTech, 200–02). Cells were kept in a humidified 5% $CO_2$ atmosphere at 37 °C.

### dCas9-KRAB transduction and selection

Primary human $CD4^+$ T cells were thawed and cultured as described above at a density of $\sim 1 \times 10^6$ cells per mL. The next day, cells were activated using Dynabeads Human T-Activator CD3/CD28 (Thermo Fisher Scientific, 11131D), at 1:1 cell:bead ratio, according to manufacturer's guidelines. Sixteen to 20 h after activation, cells were transduced with the corresponding lentiviral vector encoding a KRAB-dCas9 fusion under different promoters and a blasticidin resistance gene. Lentiviral transductions were performed in growth media supplemented with 5 mM HEPES (Thermo Fisher Scientific, 15630,080), and spinoculation was carried out at $800 \times g$ for 1 h at 37 °C. Seventy-two hours after transduction, blasticidin selection was carried out at 20 µg/mL for 2 days, followed by an additional 5 days at 12.5 µg/mL. Media was replenished, and cells were expanded as necessary based on confluency.

KRAB-dCas9 lentiviral constructs tested in this study include the following: EFS-ZIM3-dCas9-P2A-Bsd, EFS-KOX1-dCas9-P2A-Bsd, CBh-ZIM3-dCas9-P2A-Bsd, and CBh-KOX1-dCas9-P2A-Bsd (Additional file 1: Fig. S1A).

### gRNA transduction and flow cytometry analysis of CRISPRi efficiency

Following 1 day of rest after blasticidin selection, KRAB-dCas9 expressing cells were activated with Dynabeads Human T-Activator CD3/CD28 (Thermo Fisher Scientific, 11131D), at 1:1 cell:bead ratio. Sixteen to 20 h after activation, cells were transduced with a CRISPRi CROPseq lentivirus encoding gRNAs targeting either *CD4*, *CD81*, *BST2*, or *ATP1B3* or a non-targeting control (NT) gRNA, at a MOI of 0.1, in growth media supplemented with 5 mM of HEPES (Thermo Fisher Scientific, 15,630080). Spinoculation was carried out at $800 \times g$ for 1 h at 37 °C. Forty-eight hours after gRNA transduction, cells were selected with 2 µg/mL puromycin for 4 days, and selection was verified by ZsGreen expression in > 95% of the cells by flow cytometry (CytoFLEX S, Beckman Coulter).

Flow cytometry was performed at different timepoints to estimate CRISPRi efficiency by measuring protein downregulation of the target genes (*CD4*, *CD81*, *BST2*, and *ATP1B3*) compared to a NT gRNA. Briefly, ~ 100,000 cells were washed with PBS and stained for 1 h at 4 °C with an antibody targeting the corresponding protein. Next, cells were washed with PBS and analyzed by flow cytometer (CytoFLEX S, Beckman Coulter), recording 30,000–50,000 events. Antibodies used were APC anti-human *CD4* clone RPA-T4 (BioLegend, 300514), APC anti-human *CD81* clone 5A6 (BioLegend, 349510), APC anti-human *CD317* (BST2, tetherin) clone RS38E (BioLegend, 348410), and APC anti-human *CD298* clone LNH-94 (BioLegend, 341706).

### Pooled CRISPRi screen

Primary human CD4$^+$ T cells expressing CBh-ZIM3-dCas9-Bsd (transduced and selected as described above) were activated with Dynabeads Human T-Activator CD3/CD28 (Thermo Fisher Scientific, 11131D) at 1:1 cell:bead ratio, and, 16–20 h later, two million cells were transduced with the pooled CRISPRi CROPseq gRNA library (described above) at a < 0.3 MOI. The transduction was performed in growth media supplemented with 5 mM HEPES (Thermo Fisher Scientific, 15630080), and spinoculation was carried out at $800 \times g$ for 1 h at 37 ℃. Forty-eight hours later, cells were selected with 2 μg/mL puromycin for 4 days, and selection was verified by ZsGreen expression in > 95% of the cells by flow cytometry (CytoFLEX S, Beckman Coulter). Cells were cultured in growth media for an additional 6 days before processing for scRNA-seq.

### Preparation of 10X Genomics scRNA-seq libraries and sequencing

The pooled CRISPRi CROPseq screen was read out using the 3' 10X Genomics platform (10X Genomics, PN-1000075). The screen dataset contains two different scRNA-seq runs: the first one (eight channels of a 10X Genomics chip B, PN-1000073) was performed with fresh cells at the end of the CRISPRi experiment, 10 days post gRNA library transduction; the second one (32 channels across four 10X Genomics chips B, PN-1000073) was performed using cells from the same experiment that were cryopreserved 10 days after gRNA transduction and thawed in puromycin-containing media (2 μg/ml) 3 days before the 10X Genomics run. To remove dead cells and debris, the cell suspension was treated with Lymphoprep (Axis-Shield, 12HHS09) the day before loading the cells into the 10X Chromium Controller. Each channel from a 10X Genomics chip B (PN-1000073) was loaded with 16,000 cells into a Chromium Controller, following manufacturer's guidelines. 3′ gene expression libraries (v3) were prepared following manufacturer's instructions (10X Genomics, PN-1000075). gRNA amplicon libraries were generated from amplified cDNA following the protocol described in [100]. Gene expression and gRNA libraries were QCed by TapeStation 4200 (Agilent Technologies) and quantified by Qubit (Thermo Fisher, Q32851) and KAPA qPCR (Kapa Biosystems, KK4824). Multiplexed gene expression libraries were pooled at a 5:1 ratio with multiplexed gRNA libraries. The pool of gene expression and gRNA libraries were sequenced across two S4 and one S2 flow cells of a NovaSeq 6000 with 28 cycles for read 1, 91 cycles for read 2, and 8 cycles for i7 index.

### CROP-seq data processing

Sequencing data were processed using cellranger count v4.0 with default parameters. We used the human reference genome provided by 10X Genomics (hg38, Ensembl annotation version 98; https://cf.10xgenomics.com/supp/cell-exp/refdata-gex-GRCh38-2020-A.tar.gz), supplemented with artificial chromosomes containing the sequences for all gRNAs present in the library to enable recovery of gRNA-derived transcripts from the cDNA library. Additionally, a feature reference file containing all gRNA sequences present in the library was supplied to quantify UMI counts for each gRNA in the gRNA library. The barcode-rank plots for all samples were concordant with good-quality data and appropriate distinction of cell containing from empty droplets. A median of 7,218

cells (s.d. 1,445 cells) were recovered from each technical replicate. For downstream analyses, the filtered count matrices produced by cellranger were imported into R using the dropletUtils package [101, 102].

### Quality control

To remove cells of poor quality, we used scater [103] to compute QC metrics. For each barcode, we assessed the total UMI counts, total number of genes detected, and the proportion of counts mapping to mitochondrial genes (Additional file 1: Fig. S2A). We removed any barcodes that deviated by more than three median absolute deviations (MAD) from the median for any of the three metrics' distributions. To account for slight differences in sequencing depth between 10X chips, thresholds were defined independently for each sample (Additional file 1: Fig. S2A). Overall, we removed 4.07% of the barcodes. We further identified outlier barcodes with very small fraction of mitochondrial reads (deviating by more than three MADs), as these are likely to correspond to stripped nuclei, instead of cells. We confirmed that these barcodes had a substantially lower number of genes detected and were thus removed from downstream analyses.

### Doublet detection

To identify putative doublets, we used the method implemented in the scDblFinder package [104] with default parameters. Barcodes identified as doublets had higher total UMI counts, number of detected genes, and were more likely to have more than one gRNA and were thus discarded. A total of 250,195 barcodes were retained, representing singlet, good-quality cells.

### gRNA assignment

To determine which gRNAs were present in each cell, we applied a binomial test to the UMI counts obtained from the gRNA library (see above). The probability of success of each gRNA was determined from the initial representation of each gRNA in the lentiviral library, assessed by DNA sequencing. The binomial test takes into account the total library size of each cell, together with the expected proportions of each gRNA, to determine the minimum UMI count required to consider a gRNA present above background noise levels. Any gRNAs with a Bonferroni-adjusted *p*-value smaller than 0.001 were considered present in that cell. We additionally discarded any significant gRNA assignments that were supported by 3 UMI counts or fewer. Overall, we were able to confidently assign gRNA calls to 152,403 cells (60.91%). From these, almost all (95.62%) had a single gRNA assigned, consistent with experiments carried out at low multiplicity of infection.

### Normalization and batch effects assessment

Gene expression counts were normalized using the deconvolution method implemented in scran [105]. Highly variable genes were inferred with the modelGeneVar function, and the top 2000 most variable genes were used for dimensionality reduction (PCA followed by UMAP, as implemented in scater). Data visualization indicated strong separation of the two experiments performed. Thus, the experiment was included as a covariate in all differential expression analyses.

### Differential expression analysis

To determine the effects from each perturbation, we used MAST [65] to test for gene expression differences between all cells containing a particular gRNA compared to a set of randomly selected 5,000 cells containing only a non-targeting gRNA (referred to as NT cells). We restricted the analysis to genes that were detected in at least 5% of cells (10,047 genes). The model fit was done using all cells containing any of the four gRNAs for the same target, plus the background set of NT cells. We added as covariates the experiment of each sample and the detection rate as defined by MAST's authors. Then, the fitted model was used to test the effect of each gRNA, by specifying the corresponding contrast. Results were filtered to include only genes that fall within 1 Mb up/downstream of the target (defined from Ensembl's v98 annotation, with the BiomaRt package [106, 107]). *p*-values were adjusted for multiple testing with the Benjamini–Hochberg method.

To assess *p*-value calibration, we followed the same strategy as Barry et al. (2021) [68]. Briefly, the 35 non-targeting (NT) gRNAs were grouped into nine groups of four, to emulate the library structure of targeting perturbations. MAST was run in the same way as described above. Since NT gRNAs do not have a genomic location, we considered all genes tested against any targeting gRNA (i.e., any gene within 1 Mb of a targeted locus). The *p*-values reported by MAST were compared to expected *p*-values under the null (Additional file 1: Fig. S3B).

The same strategy was used to test for changes in expression for both targeting and non-targeting gRNAs using limma-voom [67] or SCEPTRE [68]. For the Wilcoxon ranksum test, we used the implementation from the pairwiseWilcox function from the scran package [105], blocking on the experiment of each sample to account for the observed batch effect.

### Target-level perturbation effects

To integrate the results from the four gRNAs targeting the same genomic locus, we used Fisher's method (as implemented in the combineGroupedPValues function from the metapod package [108]) to integrate the raw *p*-values from all four gRNAs into a single *p*-value. These target-level *p*-values were corrected for multiple testing to account for all the tests performed for the complete library (considering genes within 1 Mb of the target, accounting for 1,336 tests), using the Benjamini–Hochberg method. Genes with an adjusted target-level *p*-value < 0.05 were considered significantly differentially expressed (DEGs). DEGs were assigned to confidence tiers depending on the number of raw gRNA-level *p*-value < 0.05 with concordant effects. These are: high-confidence tier (3 or 4 gRNAs), medium-confidence tier (2 gRNAs), and low-confidence tier (1 gRNA). Additional file 3: Table S2 includes all significant E2G links identified by MAST analysis.

### CRISPR-deletions, editing assessment, and bulk RNA-seq

Primary human CD4$^+$ T cells from two independent donors were activated using Dynabeads Human T-Activator CD3/CD28 (Thermo Fisher Scientific, 11131D) at 1:1 cell:bead ratio. Seventy-two hours after activation, cells were electroporated with gRNA/Cas9-nuclease RNP complexes targeting the enhancer, in independent

reactions for each element. For elements smaller than 400 bp (HPCAL ENH and PKM ENH), three gRNAs were used (one in each flanking region and one inside the enhancer), and for elements larger than 400 bp (CTSC ENH1 and KCNN4 ENH), four gRNAs were used (one in each flanking region and two inside the enhancer) (see Table below). gRNAs were designed using the Broad Institute platform *CRISPick* [97, 109]. To electroporate the cells, (1) RNP complexes were prepared with 143 pmol of gRNAs (synthesized by IDT) and 47 pmol of Cas9-nuclease (IDT, 1081059) in duplex buffer (IDT, 1072570) and incubated at room temperature for 5–10 min; (2) 2 million cells per reaction were washed with PBS, Dynabeads were removed and cells were resuspended in P3 nucleofector solution (Lonza, V4XP-3024); (3) cells resuspended in P3 buffer were mixed with RNP complexes and electroporated using the EH-115 program of the Amaxa 4D Nucleofector X Unit (Lonza, AAF-1002X); and (4) electroporated cells were recovered in T cell media and returned to the incubator. After culturing and expanding electroporated cells for 10 days, 0.5–1 million cell pellets were harvested for genomic DNA and RNA extraction.

Genomic DNA was extracted using QuickExtract lysis buffer (Qiagen, SS000035-P) following manufacturer's recommendations. PCRs were performed for each sample using the corresponding primers to amplify the edited region (see table below) and AmpliTaq Gold Mastermix (Thermo Fisher, 10289234). PCR products were analyzed by automated electrophoresis using TapeStation 4200 High Sensitivity D5000 reagents ScreenTape system (Agilent Technologies) (Fig. 5C).

| Enhancer | gRNA sequences | Primers for gDNA PCR |
|---|---|---|
| CTSC ENH1 | 1: TTTATTACTACTAAACTGAG<br>2: AAGAGAAACTGACTTAGGTA<br>3: GGGCTTTCTCAATGACCCAA<br>4: CACCACCTATAAAGATGCTA | F: TCTCACCTTAAAGAGCTGTTGT<br>R: CGCGTATTTTGTTACAGTTCTC |
| HPCAL1 ENH | 1: GGGCTTCAACAAAGGAATTG<br>2: ACACTTCCTGGATGAGCCAT<br>3: TCCATCTACAGATTTGAGGC | F: GCATGGAGAGGGAGAAAGATTT<br>R: TGACGCTGACTTAGGGTAGAG |
| PKM ENH | 1: TGGGGTAGGAGGGCTCTACA<br>2: GATGTGGCCATCCATTGGGG<br>3: AACGGAAGGTTAAACTCCAG | F: GAGGAGAGGTCTGATGCATTTG<br>R: TGATACAGGCATGGAATGAACA |
| KCNN4 ENG | 1: CTGGACTGCTGGTCTGAGGG<br>2: GAAAGAACCCAGGTGCCTCG<br>3: CAAGGTCCCAGAGATGGCGG<br>4: CCAGGCACTGCTCAAGGAGT | F: TGGGTCTGAAGGAGGAGGAT<br>R: ACTGAGAGCAAAGAAGAGACTG |

Total RNA was extracted from frozen cell pellet samples using RNeasy Plus Mini kit (Quiagen, 74134) following manufacturer's instructions. RNA samples were quantified using Qubit 2.0 Fluorometer (ThermoFisher Scientific), and RNA integrity was checked using TapeStation 4200 (Agilent Technologies). Strand-specific RNA-sequencing libraries were prepared using NEBNext Ultra II Directional RNA Library Prep Kit for Illumina following manufacturer's instructions (NEB, E7760L). The sequencing library was validated on TapeStation 4200 (Agilent Technologies) and quantified using Qubit 2.0 Fluorometer (ThermoFisher Scientific) and quantitative PCR (KAPA Biosystems). The RNA sequencing libraries were sequenced on the Illumina NovaSeq instrument according to manufacturer's instructions, using a $2 \times 150$-bp paired end (PE) configuration.

## Bulk RNA-seq data processing and differential expression analysis

RNA-seq paired-end fragments were trimmed with Trimmomatic v0.39 [110] using the Illumina adapter sequences ILLUMINACLIP:adapters.fa:2:30:10 and the options SLIDINGWINDOW:4:15 MINLEN:36. Trimmed fragments were aligned to the human reference genome (GRCh38) using STAR [111] with options –alignSJoverhangMin 8 –outSAMunmapped Within –outSAMattributes NH HI AS NM MD –outSAMstrand-Field intronMotif. Mapped fragments were quantified using featureCounts [112] and Ensembl's genome annotation version 96 (http://apr2019.archive.ensembl.org/index.html).

To identify significantly differentially expressed genes in perturbed cells, we used edgeR 3.34.0 [113, 114]. We removed lowly expressed genes with the filterByExpr function and considered 16,752 genes in downstream analyses. Dispersion was estimated with the estimateDisp function, and model fit was performed with glmQLFit, setting robust to TRUE. For each perturbed element, we tested for changes in expression against the NT controls with the glmQLFTest function. *p*-values were corrected for multiple testing with the Benjamini–Hochberg method. A gene is considered significantly differentially expressed if its adjusted *p*-value is smaller than 0.05. Results from the differential expression analysis are available in Additional file 4: Table S3. For plotting, normalized expression estimates were computed with the cpm function having prior calculated normalisation factors with the TMM method (calcNormFactors).

## GWAS credible set intersections

For a wide range of publicly available GWAS, we generated conditional test statistics [115, 116] and conditional 95% credible sets from GWAS summary statistics using an established Bayesian fine-mapping approach [117, 118]. For each GWAS signal which overlapped the perturbed target region $+/-1$ kb, we also checked for colocalization of eQTL signals for the DEGs using the coloc software with default parameters [10, 119].

## Supplementary Information

**Additional file 1: Fig. S1.** A) Schematic of dCas9-KRAB lentiviral constructs tested. B) Bar plot showing the percentage of cells retaining protein expression for different target genes (CD4, CD81, BST2, ATP1B3) 10 days after TSS-targeting gRNA transduction into primary CD4+ T cells expressing different dCas9-repressor constructs, analysed by flow cytometry and normalised to the corresponding non-targeting gRNA control sample. gRNA #1 and #2 refer to two different gRNA designs for a given TSS. C) Bar plots for the same experiment as B), showing flow cytometry data for days 6, 10 and 16 post-gRNA transduction. D) Normalised expression levels of ATP1B3, measured by 10X Genomics 3'scRNA-seq 11 days after the corresponding targeting (red) or non-targeting (grey) gRNAs were transduced into primary CD4+ T cells expressing a CBh-ZIM3-dCas9 repressor construct. Dashed line indicates median expression level in cells with non-targeting controls. E) Percentage of cells showing protein downregulation of the target gene after CRISPRi by flow cytometry (x-axis) versus significance (-log10 FDR) of downregulation of the target gene (mRNA) by 3'scRNA-seq analysis (y-axis), normalised to the corresponding NT control. Pearson r2 = 0.98. **Fig. S2.** A) Distributions of the total UMIs per cell, number of detected genes per cell, and fraction of reads mapping to mitochondrial genes, used for quality control of the scRNA-seq data. Each violin corresponds to a technical replicate (channel in a 10X chip); colours indicate different 10X chips. The dotted lines indicate the thresholds used for each replicate to exclude poor-quality cells. B) Same as Fig. 2C but split per technical replicate. C) Barplot showing the number of cells where the same gRNA is the most abundant in both the cDNA and gRNA libraries (concordant, blue); the most abundant gRNA is different between libraries (discordant, orange); or no gRNA information was recovered from the cDNA library (yellow) D) Scatter plot of the relative abundance of each gRNA in the plasmid library (x-axis, assessed by DNA sequencing of the library) versus the number of cells positive for each gRNA (y-axis, assessed from the scRNA-seq data). **Fig. S3.** A) Barplots of the fraction of significant differentially expressed genes (DEGs) from positive control perturbations that are supported by different numbers of gRNAs (raw gRNA-level

*p*-value < 0.05). The expected genes are shown separately from all other DEGs. B) QQ plot of the expected vs observed p-values reported by MAST when testing for expression changes from non-targeting gRNAs, which should not induce any significant changes in expression. Deviation from the diagonal indicates inflated p-values. C) Same as B) but for results from SCEPTRE, limma-voom or a Wilcoxon rank sum test. D) Upset plot indicating the number of DEGs for positive control targets that are identified by any of the four methods. The height of the bar indicates the number of DEGs, split by target class (indicated by different colours). Under each bar, the methods that called the gene as significant are indicated. The expected genes are shown separately from all other DEGs. E) Same as D) but with the colours indicating the number of gRNAs supporting each DEG (raw gRNA-level *p*-value < 0.05).

**Additional file 2: Table S1.** Target selecion and gRNA library information. Coordinates are based on the human reference genome (GRCh38). For positive control targets, the expected gene is indicated.

**Additional file 3: Table S2.** MAST differential expression results for all genes within 1Mb of a perturbation and detected in at least 5% of cells. Results are aggregated at the target level.

**Additional file 4: Table S3.** Differential gene expression analysis of bulk RNA-seq data from the E2G validation experiments.

**Additional file 5.** Review history.

### Acknowledgements
We would like to thank Valeriia Sherina and Violet Zhang for insightful discussions. This study makes use of data generated by the Blueprint Consortium: a full list of the investigators who contributed to the generation of the data is available from www.blueprint-epigenome.eu. This research has been conducted using data from UK Biobank www.ukbiobank.ac.uk, a major biomedical database.

### Review history
The review history is available as Additional file 5.

### Peer-review information

### Authors' contributions
RR, CA-C, and XI-S conceived and designed the study, interpreted results, and wrote the manuscript. JE-G, AC, AA, and RR selected the elements for screening. CA-C performed all experiments. XI-S conceived and performed all computational and statistical analysis. CF and BS prepared lentiviral productions and supported the screen and validation experiments. DC and AH performed GWAS and eQTL analysis. AK conceived the gRNA calling method. WP and SU processed and analyzed bulk RNA-seq data. SM and CF supported library preparation and next-generation sequencing. RR and GD supervised the study. All authors approved the manuscript.

### Funding
This study has been funded by GSK.

### Availability of data and materials
Raw data are available in the BioStudies database (https://www.ebi.ac.uk/biostudies/) under accession number E-MTAB-13324 [120]. All code used for data processing and analysis is available at https://github.com/xibarrasoria/Tcell_crisprQTL/ [121] and at Zenodo (https://zenodo.org/doi/10.5281/zenodo.10455977) [122] under an MIT license.

## Declarations

### Ethics approval and consent to participate
The human biological samples in this study were sourced ethically, and their research use was in accord with the terms of the informed consent under an IRB-/EC-approved protocol (IRB approval number 20190318) and reviewed by the WIRB (Western Institutional Review Board). All experimental methods comply with the Helsinki Declaration.

### Consent for publication
Not applicable.

### Competing interests
C. A.-C., X. I.-S., C. F., J. E.-G., D. H., A. H., B. S., W. P., S. U., A. C., A. A., S. M., C. F., G. D., and R. R. are employees at GlaxoSmith-Kline. A. K. is an employee at Myllia Biotechnology.

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

## 