## [**Additional file 5. **Review history. · Genome Biology]

Review History

First round of review

Reviewer 1

Are you able to assess all statistics in the manuscript, including the appropriateness of statistical tests used? Yes, and I have assessed the statistics in my report.

Comments to author:

Catalinas et al. present a novel method for primary cell enabled CRISPRi screening of noncoding regulatory elements with single cell readout. They successfully applied this method to primary T cells, identifying targets for 45 noncoding regulatory elements and 35 gene promoters by profiling a substantial population of 250,000 CD4+ T cells. The authors systematically analyzed and validated the element-to-gene (E2G) relationships using QTL data and previous evidence. This method holds promise for advancing the field of gene regulation and addressing the longstanding variant to function (V2F) problem. However, there are concerns and suggestions for improvement that need to be addressed.

Many distal elements and gene promoters were chosen for CRISPRi experiments in CD4+ T cells to explore the element-to-gene (E2G) connections and potential impacts. However, it is crucial for the authors to provide a clearer and more explicit explanation regarding the rationale behind selecting these specific genomic elements. Furthermore, I have noticed the absence of negative control elements, such as inactive DNA sequences, in the primary T cells. Including such negative controls would strengthen the experimental design and help establish a more comprehensive understanding of the observed effects.

Considering the variant within different cellular contexts is of utmost importance, especially considering that a substantial portion of variants are noncoding and likely play cell type- or state-specific regulatory roles. One crucial advancement offered by single cell techniques is the ability to investigate the heterogeneity of cell states. In the study, Catalinas et al. conducted their experiments on primary T cells at a large scale, and it is essential to examine the heterogeneity within these cells. Additionally, it remains unknown whether perturbing noncoding elements affects target genes in specific cell states alone, making it imperative to explore this aspect further. By addressing these points, a more comprehensive understanding of the regulatory mechanisms and the impact of noncoding elements on target genes can be achieved.

In the analyses of perturbation effects, the authors observed a large fraction dropout that is common in single cell genomics data. This phenomenon can significantly impede the accurate interpretation of the vast cellular-level data, as demonstrated in the manuscript. However, an effective approach to mitigate this challenge is to model the cellular phenotype using all available molecular signals. By doing so, more reliable discoveries regarding perturbation effects can be achieved. To enhance the robustness of their analyses and findings, I suggest that the authors consider incorporating a well-established network propagation method, as proposed by Yu et al. in their study published in Nature Biotechnology in 2022. This addition would bolster the reliability and validity of the analyses performed by Catalinas et al.

It is crucial to explore the generalizability of the presented method to other cell types and systems. Understanding the limitations and potential challenges of implementing the method in different cellular contexts or solid tissues is essential for researchers seeking to apply it in their own studies. Investigating the feasibility of applying this method to solid tissues would be valuable. The authors should discuss any caveats, potential issues, and future directions associated with these aspects.

Reviewer 2

Are you able to assess all statistics in the manuscript, including the appropriateness of statistical tests used? Yes, and I have assessed the statistics in my report.

Comments to author:

This manuscript by Catalinas et al adapts the prior criprQTL approach from Shendure and colleagues to primary T cells, connecting disease-associated noncoding regulatory elements to protein-coding genes using pooled CRISPRi screening and single-cell RNA-seq. First, they selected an efficient CRISPRi construct and validated their experimental setup using TSS-targeting guide RNAs.

Subsequently, they designed a pool of 320 gRNAs targeting 35 TSSs, 28 enhancers previously identified in K562 cells, 3 locus control regions for CD2, and 14 ENCODE cCREs in the CRISPRi library. The first three categories served as positive controls, while 35 non-targeting gRNAs were included as negative controls. By linking the differentially expressed genes with the perturbed non-coding elements, the study identified both known and novel element-to-gene pairs. Furthermore, several of these non-coding regulatory elements also overlapped with disease-associated SNPs and immune-related traits. In summary, this study proposed a CRISPRi-based single-cell RNA-seq approach and developed an analytical pipeline for mapping the non-coding regulatory elements to the effector genes, offering a complementary methodology to GWAS and eQTL analyses.

A key strength of this paper is that it links non-coding regulatory elements, disease-associated variants, and effector genes at a single-cell resolution in primary cells. These findings complement immune-related element-to-effector gene connections where no eQTL evidence exists. However, several aspects of the study require further clarification.

My major comments are:

1. The comparison in Fig 1B-E is sloppy. All targets should be assessed with at least 3 guides and preferably more (the authors use only 1 or 2 guides per target, which is not the standard in the field). One construct (SFFV) varies the position of the effector and the promoter but does not control properly for varying each of these independently. Please test the position of the effector with all promoters or remove the SFFV comparison. The summary plot needs to have error bars and significance testing before making any strong claims about better and worse effectors.
2. The MeCP2-KOX1 double-effector should be tested with the smaller promoters (e.g. EFS, CBh), given that the SFFV promoter also performs the worst with the single effector construct.

3. In Figure 2C, there is a reported 39.1% of cells with unassigned gRNAs. Given that cells underwent approximately 10 days of puromycin selection post-gRNA lentiviral transduction, it is assumed that all cells used in the 10x Genomics should have processed at least one gRNA. Further elaboration is needed to explain the specific cell population used for the pie chart.
4. Please validate a subset of the novel Element-to-Gene identified in this study. For example, you can individually inhibit the three non-coding elements associated with SNPs and disease shown in Figure 5 using CRISPRi. Subsequently, the impact on linked gene expression could be measured via FACS, qPCR, and/or western blot.
5. In the "Evaluation of Analytical Methods for High Confidence cis E2G Mapping using criprQTL" section, you observed inflated P-values from four algorithms used to define differentially expressed genes. Consequently, they opted for MAST combined with a target-level P-value from the Fisher test. Please provide QQ plots similar to Supplementary Figures S3B for the Fisher test P-values.
6. For Figures 5B and 5F, please provide the effect size information in the violin plot. Additionally, clarity is needed regarding the single cell number with the 35 non-targeting gRNAs (if you merge them for the NT group) and the target group for the violin plots on the left. Considering the use of corrected P-values derived from MAST outputs, further explanation is needed on the low FDR value on the top of the panel due to its potential connection to the original inflated MAST P-values. Furthermore, please add statistical tests for the bar plots on the right side of both panels.
7. In Figures 5I-J, a novel E2G link is identified that hasn't been previously reported. In addition to this novel link, do you observe any SNPs or disease traits overlap with the other 9 intron cCREs and two intergenic cCREs linked to significant DEG?
8. For the Fig. 1F, please provide the single cell numbers expressing each gRNAs in the violin plots.

Minor comments:

1. In the "Lentiviral Vector Production and Determination of Viral Titre" section, it's noted that viral titre assay was conducted in HEK cells. Given the cell-line specificity of virus transduction efficiency, clarification is needed on whether the titre assay was also performed in CD4+ T cells to optimize the virus volume for low MOI.

Reviewer #1:

Catalinas et al. present a novel method for primary cell enabled CRISPRi screening of noncoding regulatory elements with single cell readout. They successfully applied this method to primary T cells, identifying targets for 45 noncoding regulatory elements and 35 gene promoters by profiling a substantial population of 250,000 CD4+ T cells. The authors systematically analyzed and validated the element-to-gene (E2G) relationships using QTL data and previous evidence. This method holds promise for advancing the field of gene regulation and addressing the longstanding variant to function (V2F) problem. However, there are concerns and suggestions for improvement that need to be addressed.

We appreciate the reviewer's generous comments regarding our E2G mapping methodology and the manuscript.

1. Many distal elements and gene promoters were chosen for CRISPRi experiments in CD4+ T cells to explore the element-to-gene (E2G) connections and potential impacts. However, it is crucial for the authors to provide a clearer and more explicit explanation regarding the rationale behind selecting these specific genomic elements.

We appreciate the reviewers concerns and have amended the Methods section to include further details about the genomic element selection. Furthermore, the full list of TSSs, LCRs and distal elements is provided in Table S1, including the gRNA sequences used and their genomic coordinates. We provide an excerpt below and highlight additions in yellow.

"The 35 TSSs were selected based on the following criteria (Table S1): 1) 15 genes with essential roles in T cell regulation; and 2) 20 genes selected based on expression, as assessed from a scRNA-seq dataset from CD3/CD28 stimulated CD4+ T cells (Cano-Gamez et al. 2020) (genes were binned into quartiles based on average expression: 11 highly-expressed genes were selected from the top quartile, 6 from the 3rd expression quartile and 3 from the lowest two expression quartiles).

The genomic coordinates for the three CD2 LCRs were obtained from (Lake et al. 1990, Festenstein et al. 1996, Kaptein et al. 1998) (Table S1).

We selected enhancers from the set of 470 high-confidence intergenic E2G pairs from Gasperini et al. (2019) (Gasperini et al. 2019) with the following features: 1) the putative enhancer must overlap a region of open chromatin in primary CD4+ T cells as identified in at least one of the following datasets: i) ATAC-seq peaks identified on CD4+ T naïve or memory cells after 16 hours or 5 days of CD3/CD28 activated stimulation (Soskic et al. 2019) or ii) ATAC-seq peaks from multiple CD4+ T cells from (Calderon et al. 2019); 2) the expected target gene, as defined by Gasperini et al. (Gasperini et al. 2019), was amongst the top 50% expressed genes in primary T cells, as assessed from a scRNA-seq dataset from CD3/CD28 stimulated CD4+ T cells (Cano-Gamez et al. 2020). After applying these filters, a set of 28 E2G pairs were selected (Table S1).

Additionally, we selected 11 intronic and 3 intergenic cCREs (ENCODE-Project-Consortium 2012) overlapping open chromatin regions in primary T cells (Calderon et al. 2019) (Table S1)."

2. Furthermore, I have noticed the absence of negative control elements, such as inactive DNA sequences, in the primary T cells. Including such negative controls would strengthen the experimental design and help establish a more comprehensive understanding of the observed effects.

We have included 35 non-targeting gRNAs to serve as negative controls. In a previous experiment in the Jurkat (T cell line) and Beas2B (bronchial epithelial cell line) cell lines, we have extensively compared non-targeting control gRNAs with gRNAs targeting inactive DNA sequences (overlapping open chromatin). We targeted 22 such inactive DNA regions in each cell line, with up to 10 different gRNAs each. Only 2 gRNAs in the Jurkat cell line had a significant effect on the expression of genes within 1Mb of the targeted loci; and no significant results were observed for any of the gRNAs in Beas2Bs (please see figure below). Therefore, in the current experiment, acknowledging the complexities of working with primary T cells, we chose to exclude control gRNAs targeting inactive DNA sequences.

One could have also targeted heterochromatin or known safe harbour sites. However, the former option is confounded by the variable/reduced ability of the Cas9 enzyme to penetrate closed chromatin whilst the latter option was not extensively explored by the field or us at the time.

3. Considering the variant within different cellular contexts is of utmost importance, especially considering that a substantial portion of variants are noncoding and likely play cell type- or state-specific regulatory roles. One crucial advancement offered by single cell techniques is the ability to investigate the heterogeneity of cell states. In the study, Catalinas et al. conducted their experiments on primary T cells at a large scale, and it is essential to examine the heterogeneity within these cells. Additionally, it remains unknown whether perturbing noncoding elements affects target genes in specific cell states alone, making it imperative to explore this aspect further. By addressing these points, a more comprehensive understanding of the regulatory mechanisms and the impact of noncoding elements on target genes can be achieved.

We thank the reviewer for highlighting the importance of single-cell variant-to-gene mapping techniques in uncovering cell state specific regulatory mechanisms. In the present manuscript, we defined a scalable protocol that would enable such studies and address this aspect in the discussion. However, the size of this experiment and the plastic/fluid nature of “steady state” T cells precludes such analysis, since the number of cells in each state with a particular gRNA is too low for a meaningful differential expression analysis.

Nevertheless, we have explored the transcriptional heterogeneity in our dataset and observe three major clusters. We can further delineate broad Th1, Th2, Th17 and Tregs subtypes based on expression of key lineage markers (e.g. TBX21, GATA3, RORA, FOXP3, etc). However, in our experience we find

that subpopulation mapping is more meaningful, discrete and robust when cells (derived from multiple donors) are cultured in the presence of lineage specific cytokines.

In a recent, distinct study (manuscript in early stages of preparation) we have explored the effects of element and gene perturbations in primary T cells (derived from 4 donors), in the presence of Treg inducing cytokines. In this study, we achieve the cell numbers required to perform differential expression analysis across substates.

4. In the analyses of perturbation effects, the authors observed a large fraction dropout that is common in single cell genomics data. This phenomenon can significantly impede the accurate interpretation of the vast cellular-level data, as demonstrated in the manuscript. However, an effective approach to mitigate this challenge is to model the cellular phenotype using all available molecular signals. By doing so, more reliable discoveries regarding perturbation effects can be achieved. To enhance the robustness of their analyses and findings, I suggest that the authors consider incorporating a well-established network propagation method, as proposed by Yu et al in their study published in Nature Biotechnology in 2022 (Yu et al. 2022). This addition would bolster the reliability and validity of the analyses performed by Catalinas et al.

We agree that data sparsity is a challenge of scRNA-seq data and welcome the reviewer's suggestion. However, we deem that the network propagation method developed by Yu et al. is not suitable in our context as it relies on a kNN graph built from the single-cell profiles. Since the perturbation effects of regulatory elements are subtle, we do not observe shifts in the (overall) transcriptional state of the perturbed cells, and thus the connectivity captured in the graph is not related to perturbation status. Instead, we chose to use MAST to directly assess the changes in expression between perturbed and NT-control cells. This algorithm has been specifically designed for scRNA-seq data and includes explicit modelling of the dropout rates for each gene to account for data sparsity.

5. It is crucial to explore the generalizability of the presented method to other cell types and systems. Understanding the limitations and potential challenges of implementing the method in different cellular contexts or solid tissues is essential for researchers seeking to apply it in their own studies. Investigating the feasibility of applying this method to solid tissues would be valuable. The authors should discuss any caveats, potential issues, and future directions associated with these aspects.

We thank the reviewer for this suggestion and have added the text highlighted in yellow below to the discussion:

“To enable the perturbation of regulatory elements, we established CRISPRi in primary CD4+ T cells by lentiviral transduction of both dCas9-repressor and pooled gRNA library. A similar approach (Schmidt et al. 2022) was used for gene silencing in primary T cells using the KOX1 KRAB domain, to enable fluorescence-activated cell sorting-based screens. We envision that our CRISPRi-based E2G methodology can be applied to other cell types and tissues that are amenable to viral transduction. However, this will most likely require extensive optimisation of transduction conditions, including testing of viral pseudotypes that can facilitate efficient infection of the desired cell type. Similarly, we found that gRNA detection in scRNA-seq libraries from primary T cells can be particularly challenging and thus, we anticipate this will require systematic optimisation in other primary cell models.”

Reviewer #2: Review comments

This manuscript by Catalinas et al adapts the prior crisprQTL approach from Shendure and colleagues to primary T cells, connecting disease-associated noncoding regulatory elements to protein-coding genes using pooled CRISPRi screening and single-cell RNA-seq. First, they selected an efficient CRISPRi construct and validated their experimental setup using TSS-targeting guide RNAs.

Subsequently, they designed a pool of 320 gRNAs targeting 35 TSSs, 28 enhancers previously identified in K562 cells, 3 locus control regions for CD2, and 14 ENCODE cCREs in the CRISPRi library. The first three categories served as positive controls, while 35 non-targeting gRNAs were included as negative controls. By linking the differentially expressed genes with the perturbed non-coding elements, the study identified both known and novel element-to-gene pairs. Furthermore, several of these non-coding regulatory elements also overlapped with disease-associated SNPs and immune-related traits. In summary, this study proposed a CRISPRi-based single-cell RNA-seq approach and developed an analytical pipeline for mapping the non-coding regulatory elements to the effector genes, offering a complementary methodology to GWAS and eQTL analyses.

A key strength of this paper is that it links non-coding regulatory elements, disease-associated variants, and effector genes at a single-cell resolution in primary cells. These findings complement immune-related element-to-effector gene connections where no eQTL evidence exists. However, several aspects of the study require further clarification.

We thank the reviewer for highlighting the advancements brought by our manuscript and, particularly, the importance of developing functional experimental pipelines in primary cells for the validation of GWAS hits. We agree that the combination of experimental and QTL-based techniques are likely to help close the colocalization gap.

My major comments are:

1. The comparison in Fig 1B-E is sloppy. All targets should be assessed with at least 3 guides and preferably more (the authors use only 1 or 2 guides per target, which is not the standard in the field). One construct (SFFV) varies the position of the effector and the promoter but does not control properly for varying each of these independently. Please test the position of the effector with all promoters or remove the SFFV comparison. The summary plot needs to have error bars and significance testing before making any strong claims about better and worse effectors.

We recognise the reviewer's concern and have addressed this by performing new experiments and revising Fig 1 and the corresponding text in the results section (highlighted in blue in the main manuscript). We also offer a brief explanation below:

- For panel 1B-E (original manuscript; Fig1B-D and Fig S1B-E in revised manuscript), we selected the top 1-2 guides targeting the selected genes and observed robust silencing efficiency. These guides have been preselected and extensively tested in our group across multiple projects/cell systems. Therefore, given the challenges with procuring, culturing and transducing primary T cells, we saw little need in testing further guides for the initial method development stage. Generally, these limitations imply that others in the field have also opted to use 1-2 gRNAs to perform CRISPR experiments in primary cells (e.g. (Schmidt et al. 2022)). Nevertheless, we fully appreciate the reviewer's concern around the robustness of gRNA effects and hence, importantly, our crisprQTL screen was performed with 4 gRNAs per target.

- We agree that we are not properly controlling for potential KRAB position effects in the SFFV construct and, as such, we have removed this construct from the figure and text.
- The summary plot comparing all dCas9 constructs across multiple genes (Fig. 1D in original submission) has been moved to Fig. S1B since this experiment was performed in a single donor. We agree that testing in multiple donors should have been carried out in order to support strong claims on the improved performance of the ZIM3 domain. Therefore, we chose to remove these statements from the main text.
- Most importantly, to address the comment about replication, we have performed a new experiment (Fig 1C in revised manuscript) where we selected three genes and performed CRISPRi across four independent primary T cell donors in a time-course, showing robust silencing efficiency.

2. The MeCP2-KOX1 double-effector should be tested with the smaller promoters (e.g. EFS, CBh), given that the SFFV promoter also performs the worst with the single effector construct.

The reviewer raised an important point, however, given the results we obtained with the ZIM3 domain and the challenges with transducing large lentiviral constructs into primary T cells, we opted to remove the mention of the MeCP2-KOX1 construct from the results and the methods.

3. In Figure 2C, there is a reported 39.1% of cells with unassigned gRNAs. Given that cells underwent approximately 10 days of puromycin selection post-gRNA lentiviral transduction, it is assumed that all cells used in the 10x Genomics should have processed at least one gRNA. Further elaboration is needed to explain the specific cell population used for the pie chart.

After gRNA transduction, cells underwent 4 days of puromycin selection and were then cultured for a further 6 days (in the absence of selection) before loading into the 10x Chromium. At this point, we confirmed that >95% of cells were positive for ZsGreen and thus, as suggested by the reviewer, the vast majority should express a gRNA. However, it is rather uncommon to detect a gRNA in all cells in scRNA-seq data.

In our case, the 39.1% of cells without an assigned gRNA are not cells *without* gRNA expression, but instead represent cells with poor gRNA transcript *recovery* in the single cell data. Most of these cells are very likely to contain a gRNA but the amount of gRNA sequencing reads recovered are not enough to assign the gRNA identity with confidence. We found this issue to be more acute in *in-vitro* cultured primary cell models, as compared to cell lines.

Importantly, we opted for a strict gRNA assignment. First, we developed a novel approach to model the signal-to-noise ratio in each cell using a binomial distribution and test whether there is enough evidence to support assigning a gRNA to a cell. Then, for cells with a significant assignment we only consider cases where the assigned gRNA is supported by at least 4 UMI counts; all cells with 3 counts or fewer are *unassigned* (please see methods section). We show a few representative examples on

the right: in the top two rows, many gRNA counts are recovered and we are able to confidently assign the gRNA(s) present in each cell. Conversely, in the 3rd row, only a few gRNA counts are recovered in these cells, and thus we cannot differentiate signal from noise with confidence.

To clarify this further in the manuscript, we highlight this aspect in the legend of figure 2C.

4. Please validate a subset of the novel Element-to-Gene identified in this study. For example, you can individually inhibit the three non-coding elements associated with SNPs and disease shown in Figure 5 using CRISPRi. Subsequently, the impact on linked gene expression could be measured via FACS, qPCR, and/or western blot.

We thank the reviewer for this comment and have performed further experiments to address this important point. We have selected four elements and targeted them using an orthogonal approach involving CRISPR-mediated targeted element deletions (in cells derived from 2 donors) and investigated the impact on gene expression via bulk RNA-seq. We describe the outcome of this experiments in the Results section and in the new Figure 5.

5. In the "Evaluation of Analytical Methods for High Confidence cis E2G Mapping using criprQTL" section, you observed inflated P-values from four algorithms used to define differentially expressed genes. Consequently, they opted for MAST combined with a target-level P-value from the Fisher test. Please provide QQ plots similar to Supplementary Figures S3B for the Fisher test P-values.

We did not include the QQ plot for Fisher p-values since this approach is not intended to alleviate p-value miscalibration. It is instead a strategy to summarise the results from independent gRNAs targeting the same locus into a single measure of significance. To minimise the rate of false discoveries we require concordant evidence from at least two different gRNAs to consider a perturbation effect in downstream analyses (confidence tiers medium and high). P-value inflation can only be addressed by including multiple biological replicates, as we highlight in the discussion.

6. For Figures 5B and 5F, please provide the effect size information in the violin plot. Additionally, clarity is needed regarding the single cell number with the 35 non-targeting gRNAs (if you merge them for the NT group) and the target group for the violin plots on the left. Considering the use of corrected P-values derived from MAST outputs, further explanation is needed on the low FDR value on the top of the panel due to its potential connection to the original inflated MAST P-values. Furthermore, please add statistical tests for the bar plots on the right side of both panels.

We have included the effect size along with the FDR for all the relevant panels in Figure 6 (originally Figure 5). We also added the number of cells in each violin and barplot. The barplots are used as an alternative representation to highlight changes in the fraction of cells with detected expression for lowly expressed genes, which is a clearer representation to violin plots when most cells have 0 counts. The results for the statistical test apply to both the violin and barplots since MAST models both the expression levels and the fraction of dropouts. Additionally, we have highlighted with * the gRNAs that are significant at the gRNA level.

7. In Figures 5I-J, a novel E2G link is identified that hasn't been previously reported. In addition to this novel link, do you observe any SNPs or disease traits overlap with the other 9 intron cCREs and two intergenic cCREs linked to significant DEG?

We observe at least one medium or high-confidence DEG for both intergenic perturbation and for all but two of the intronic perturbations.

From the intergenic elements, an element located 25kb away from *TGFB1* results in dysregulation of 5 genes within 1Mb. The largest effect is observed for *TGFB1*, with a significant downregulation of ~21% compared to NT controls, from 3 gRNAs that show an effect. The perturbed element overlaps a credible set from an osteoarthritis GWAS and we observed colocalization with an eQTL in adrenal tissue. The other intergenic element, which lies 17kb away from *SMARCE1*, results in upregulation of *TOP2A* and *RARA*. The strongest effect was a 16% increase in *TOP2A* expression levels but we did not find colocalization with eQTLs. As such, due to lack of or non-T cell relevant eQTL colocalization, we do not elaborate on these two elements in the manuscript.

From the nine intronic perturbations with at least one DEG, five result in downregulation of the overlapping gene. These effects, however, are more difficult to interpret since we cannot differentiate effects from silencing the intronic element from downstream effects due to the silencing of the overlapping gene. Perturbation of four of these elements result in additional DEGs (beyond the overlapping gene). Overall, some of these elements overlap with credible sets for relevant immune diseases, however, we did not find colocalization in relevant tissues.

A complete list of DEGs is provided in Table S2. Additionally, we provide a thorough analysis of the perturbation effects of all the elements in the github repository (https://github.com/xibarrasoria/Tcell_crisprQTL/), for people to explore in more detail.

8. For the Fig.1F, please provide the single cell numbers expressing each gRNAs in the violin plots.

We have added the number of cells as requested.

Minor comments:

1. In the "Lentiviral Vector Production and Determination of Viral Titre" section, it's noted that viral titre assay was conducted in HEK cells. Given the cell-line specificity of virus transduction efficiency, clarification is needed on whether the titre assay was also performed in CD4+ T cells to optimize the virus volume for low MOI.

Thank you for noticing this. Indeed, we performed the titration in HEK cells but also primary CD4+ T cells and selected a lentiviral dilution for low MOI. We have amended the methods section accordingly and we also show below an example of the titration of the gRNA library performed in CBH-ZIM3-dCas9 CD4+ T cells, assessed by expression of the ZsGreen reporter by flow cytometry.

Second round of review

Reviewer 1

All of my concerns have been addressed. Thank you for your effort in making revisions.